

# Measurement Report: Cloud and environmental properties associated with aggregated shallow marine cumulus and cumulus congestus

Ewan Crosbie[1,2], Luke D. Ziemba[1], Michael A. Shook[1], Taylor Shingler[1], Johnathan W. Hair[1], Armin Sorooshian[3,4], Richard A. Ferrare[1], Brian Cairns[5], Yonghoon Choi[1,2], Joshua DiGangi[1], Glenn S. Diskin[1], Chris Hostetler[1], Simon Kirschler[6,7], Richard H. Moore[1], David Painemal[1,2], Claire Robinson[1,2,*], Shane T. Seaman[1], K. Lee Thornhill[1,2], Christiane Voigt[6,7], Edward Winstead[1,2]

[1]NASA Langley Research Center, Hampton, VA 23666, U.S.A.
[2]Analytical Mechanics Associates, Inc., Hampton, VA 23666, U.S.A.
[3] Department of Chemical and Environmental Engineering, University of Arizona, Tucson, AZ 85718, U.S.A.
[4] Department of Hydrology and Atmospheric Sciences, University of Arizona, Tucson, AZ 85718, U.S.A.
[5]NASA Goddard Institute for Satellite Studies, New York, NY 10025, U.S.A.
[6]Institut für Physik der Atmosphäre, Deutsches Zentrum für Luft- und Raumfahrt (DLR), Oberpfaffenhofen, Germany
[7]Institut für Physik der Atmosphäre, Johannes Gutenberg-Universität, Mainz, Germany
* deceased

*Correspondence to*: Ewan Crosbie (ewan.c.crosbie@nasa.gov)

**Abstract.**

Mesoscale organization of marine convective clouds into linear or clustered states is prevalent across the tropical and subtropical oceans and its investigation served as a guiding focus for a series of process study flights, conducted as part of the Aerosol Cloud meTeorology Interactions oVer the western ATlantic Experiment (ACTIVATE) during summer 2020, 2021 and 2022. These select ACTIVATE flights involved a novel strategy for coordinating two aircraft, with respective remote sensing and in situ sampling payloads, to probe regions of organized shallow convection for several hours. The main purpose of this measurement report is to summarize the aircraft sampling approach, describe the characteristics and evolution of the cases, and provide an overview of the datasets that can serve as a starting point for more detailed modeling and analysis studies.

Six flights are described, involving a total of 80 dropsonde profiles that capture the environment surrounding clustered shallow convection together with detailed observations of the vertical structure of cloud systems, comprising up to 20 altitude levels that were sampled in situ. Four cases involved deepening convection rooted in the marine boundary layer that developed vertically to 2-5 km with varying precipitation amounts, while two cases captured more complex and developed cumulus congestus systems extending above 5 km. In addition to the thermodynamic and dynamic characterization afforded by dropsonde and in situ measurements, the datasets include cloud and aerosol microphysics, trace gas concentrations, aerosol and droplet composition, and cloud and aerosol remote sensing from high spectral resolution lidar and polarimetry.



## 1 Introduction

Cumulus convection is a pervasive component of the marine atmosphere above the global tropical and subtropical oceans
(Warren et al., 1988; Johnson and Lin, 1997; Bony et al., 2004) where vertical transport and overturning circulations are
tightly coupled to diabatic processes associated with radiative and latent heating (Riehl and Malkus, 1958; Johnson et al.,
1999; Sobel and Bretherton, 2000). In addition to their role in the heat, moisture, and momentum budgets, convective clouds
across all scales affect the production, loss, and vertical distribution of atmospheric trace gases (e.g., Dickerson et al., 1987;
Fried et al., 2016; Li et al., 2018) and aerosol particles (e.g., Koch et al., 2003; Berg et al., 2015; Corr et al., 2016;
Wonaschuetz et al., 2012; Reid et al., 2019; Leung and van den Heever, 2022).  Tropical cloudiness has been associated with
three major modes (Johnson et al. 1999; Haynes and Stephens, 2007): (i) a shallow cumulus mode, often capped by a trade
wind inversion, (ii) a middle mode associated with cumulus congestus and altocumulus, often associated with enhanced
stability near the melting level (Posselt et al., 2008), and (iii) a deep mode (not considered here) associated with
cumulonimbus and anvil cirrus, capped by the tropopause.

Oceanic regions that frequently accommodate deep convection are also subject to suppressed periods with shallow
convection (Johnson and Lin, 1997; Malkus and Riehl, 1964). The vertical distribution and extent of clouds in unstable
environments capable of supporting deep convection are also strongly dependant on the moisture profile (Redelsperger et al.,
2002; Jensen and Del Genio, 2006; Takayabu et al., 2006) because of the inhibiting influence of entrainment and mixing on
updraft buoyancy (Derbyshire et al., 2004).  Indeed, pre-moistening of the mid-troposphere (Kuang and Bretherton 2006;
Waite and Khouider, 2010) and upscaling of cloud-forming updrafts (e.g., through cold pools, Khairoutdinov and Randall,
2006), as well as moisture convergence (Hohenegger and Stevens, 2013), have been thought necessary to facilitate the
growth from shallow to deep convection.  For thermodynamic environments that lack pronounced stable layers (e.g., a
demarcated cloud-capping trade-wind inversion), separation of the shallow cumulus mode from growth into congestus may
be more ambiguous and resigned to a subjective altitude threshold (e.g., 3-4 km). Similarly, a subset of clouds identified or
classified as congestus may encompass nascent energetic growth of transient systems on their way to becoming deep
convection (e.g., Luo et al., 2009), which may physically differ from terminal congestus, where development has ceased
(Leung and van den Heever, 2022).

In addition to understanding factors that control the vertical distribution of cumulus clouds, there has been a surge of recent
interest in the spatial distribution of convection, particularly spurred by the propensity for deep and shallow convection to
self-aggregate in cloud resolving models under certain conditions. Simulations run close to a state of radiative-convective
equilibrium over a homogeneous ocean surface can produce deep convection that progressively self-aggregates (Held et al.,
1993; Tompkins, 2001; Tompkins and Craig, 1998; Bretherton et al., 2005; Muller and Held, 2012; Wing and Emanuel,
2014) and mimics the tendency for oceanic convection in the real tropical atmosphere to structurally organize across a wide
range of horizontal length scales (Holloway et al., 2017; Mapes and Houze, 1993; Zuidema 2003; Stein et al., 2017; Tobin et



al., 2012; Semie and Bony, 2020; Masunaga, 2014; Nesbitt et al., 2006). Large eddy simulations (LES) of shallow marine cumulus have been found to exhibit similar mesoscale self-aggregation of moisture and cloudiness, attributed purely to moisture advection and negative gross moist stability through model experiments that suppressed interactive radiation, surface fluxes, and precipitation (Bretherton and Blossey, 2017; Narenpitak et al., 2022; Janssens et al., 2023). In contrast, using LES experiments replicating conditions encountered during the Rain in Cumulus over the Ocean experiment (RICO; Rauber et al., 2007), Seifert and Heus (2013) found that cold pools from shallow cumulus precipitation were necessary to produce the cloud organization into mesoscale arcs. Furthermore, shipboard observations during RICO confirmed the presence of frequent convective showers associated with shallow cumulus and their accompanying cold pools (Zuidema et al., 2012). Linkages have been made between precipitation-mediated shallow cloud organization and the influence of aerosols in modelling studies (Xue et al., 2008; Wang et al., 2010) and observations (Wood et al., 2018; Mohrmann et al., 2019; Goren et al., 2019). A wide spectrum of shallow cloud organization types have been identified in nature ranging from patterns found in stratocumulus (e.g., Wood and Hartmann, 2006), organization in mid-latitude cold air outbreaks (Agee 1987, Atkinson and Zhang, 1996), fair weather cumulus rolls (Lemone and Meitin, 1984), cloud patterns found in deeper boundary layers of the downstream trades (Schulz et al., 2021; Denby 2020; Stevens et al., 2020; Janssens et al., 2021), and those associated with cold pools and congestus (Snodgrass et al., 2009; Zuidema et al., 2012; Rowe and Houze, 2015; Ruppert and Johnson, 2015). There is a clear need for continued observational efforts focused on cumulus aggregation to test and/or verify the findings from idealized numerical model simulations and provide detailed observational support for the driving physical and dynamical processes across these various regimes.

Shallow cumulus clouds modulate albedo and changes in their fractional coverage, vertical extent, and microphysical properties can exert a strong influence on regional and global climate (Bony et al., 2004; Vial et al., 2016; Rieck at al., 2012). In addition, the response of low-lying shallow cumulus, particularly in the trade wind regions, to climate warming constitutes a sizable uncertainty in climate model cloud feedbacks (Bony and Dufresne, 2005; Bretherton 2015; Bretherton et al., 2013; Sherwood et al., 2014; Webb and Lock 2013). The amount of cloud coverage near cloud base (i.e., resulting from the contributions of very small cumulus) dominates the overall cloud fraction (Nuijens et al., 2014). Increases in convective mixing, driven by increased mass flux in deeper, active cumulus, projected to occur in future warming (Vial et al., 2016), has been postulated to result in desiccation of neighbouring small clouds by entrainment drying (Brient and Bony, 2013; Sherwood et al., 2014; Brient et al., 2015), thus reducing the cloud base cloud fraction. However, recent analysis of trade-wind cumulus observations has cast doubt on the strength of this feedback (Vogel et al., 2022), suggesting the importance of mesoscale organization of these clouds and ubiquitous low-level mesoscale overturning circulations that drives the distribution of cloudiness (George et al., 2023). Variations in the type of mesoscale organization has been shown to result in changes in fractional coverage of shallow clouds and resultant cloud radiative effects (e.g., Bony et al., 2020), motivating the need for process-level understanding and effectual realization of their associated properties in climate models.

Microphysical properties of warm (ice free) cumulus are controlled, in part, by the availability of aerosol particles to act as cloud condensation nuclei (CCN) and by dynamics (Kirschler et al., 2022). Clouds forming in higher CCN environments



result in smaller cloud droplets for fixed liquid water content (Twomey, 1977) that may delay and/or suppress the formation of precipitation (e.g., Rosenfeld and Lensky, 1998; Khain et al., 2005) and affect how cloudy parcels interact with their environment (e.g., Xue and Feingold, 2006). Retained cloud water and changes to the condensation rate may affect updraft buoyancy in competing ways (Igel and van den Heever, 2021; Grabowski and Morrison, 2021; Fan et al., 2018), affecting both shallow clouds that remain liquid only (e.g., Koren et al., 2014) and the potential for invigoration in the ice phase (e.g.,

Rosenfeld et al., 2008). In concert, these aerosol-mediated changes may influence the timeline of cloud growth, coverage, vigor, terminal vertical extent, and lifecycle precipitation (Koren et al., 2005; Koren et al., 2008; Tao et al., 2012; Igel and van den Heever, 2021; Barthlott et al., 2022; Fan et al., 2018; Rosenfeld et al., 2008; Storer and van den Heever, 2013; Khain et al., 2005; Marinescu et al., 2021).

Lateral and vertical mixing processes involving interactions between clouds and the surrounding environment are ubiquitous

(Romps and Kuang, 2010) and exert control on the vertical distribution of cloud water (Rangno and Hobbs, 2005). Entrainment of subsaturated environmental air may influence the droplet population differently depending on the relative timescales of turbulent homogenization and microphysical response (Baker et al., 1980; Burnet and Brenguier, 2007; Jensen and Baker, 1989), which is a function of both the droplet sizes and the length scales of mixing (Kumar et al., 2018). The spatial and temporal heterogeneity of clouds, precipitation, and aerosols (and feedbacks therein) confounds efforts to

understand aerosol-cloud interactions (Gryspeerdt et al., 2015; Varble et al., 2018), while aerosol effects on cloud microphysics may be modulated by the environment (Storer et al., 2010; Sokolowsky et al., 2022). Clouds also mediate the microphysical properties of subsequent clouds through their influence on pre-existing CCN properties (Hoppel et al., 1994; Feingold and Kreidenweis, 2000), the removal of CCN by rainout (Textor et al., 2006; Wang et al., 2020; Flossman et al., 1985) and the lofting of precursor gases that nucleate new particles (Williamson et al., 2019). In summary, the myriad multi-

path interactions amongst aerosols, clouds, radiation, and meteorology from the cloud system to the droplet scale may all contribute to the complexity of mesoscale aggregation of marine cumulus and motivate the need for targeted observations and further modelling.

In this paper, we present observational case studies associated with targeted aircraft measurements of aggregated shallow cumulus and terminal cumulus congestus that were conducted over the summertime subtropical western North Atlantic near

the coastal United States and Bermuda. Although this region is situated near the latitudinal extent of the tropics, sea surface temperatures (SST) are at a range close to typical of tropical basins (298-300 K) and low-level warm, moist advection driven by flow around the subtropical ridge produces airmasses that are thermodynamically similar to conditions found in tropical maritime regions. These cases encompass a variety of mesoscale cloud conditions, exemplified by the nature of cloud organization and aggregation, vertical extent, macro- and microphysical properties, and environmental attributes, while

occurring under relatively consistent larger scale environments. The measurements include both in situ and remote sensing datasets from two coordinated aircraft platforms that specifically targeted regions of aggregated shallow convection. The main purpose of this measurement report is to summarize the aircraft sampling approach, describe the characteristics and



evolution of the cases, and provide an overview of the datasets that can serve as a starting point for more detailed modeling and analysis of this set of case studies.

## 2 Methods

### 2.1 ACTIVATE

The Aerosol Cloud meTeorology Interactions oVer the western ATlantic Experiment (ACTIVATE) was conducted from NASA Langley Research Center and Bermuda during 2020-2022, comprising six airborne measurement campaigns split between winter and summer each year (Sorooshian et al., 2019). ACTIVATE employed a unique coordinated aircraft strategy for remote sensing and in situ sampling of clouds, aerosols, and trace gases that involved speed matching a turboprop King Air B200 or UC12 (King Air) at high altitude with a low-flying Dassault HU-25 Falcon jet (Falcon) allowing both aircraft to remain horizontally collocated. Most flights used the coordinated aircraft in a survey pattern to build statistics (e.g., Kirschler et al., 2023), while a minority of flights were assigned to process studies during each season. During summer, process study flights were used specifically to probe organized regions of aggregated shallow cumulus and cumulus congestus.

Unlike statistical surveys where both aircraft prioritized following a single path, summer process study flights prescribed separate, but coordinated, patterns for each aircraft that were anchored to a target cloud system or convective feature (Figure 1). At a nominal cruise altitude of 9 km, the King Air flew at least five transects across the target on different azimuths connected by shorter perimeter legs. Each transect covered nominally 80 km with dropsondes released near the start and end of each transect to create a perimeter, as well as occasional placements near the center. Meanwhile, the Falcon performed a series of short, constant altitude penetrations of the target cloud system and the near-field environment, repeated at multiple levels that also included a leg just above the highest cloud top and at least one leg below the lowest cloud base. Legs that penetrated cloud began close to cloud top, where there was typically a single emergent convective core, then progressively moved down in altitude, usually resulting in longer sampling legs as the cloudy region expanded to involve multiple cores, as illustrated schematically in Figure 1a. The Falcon sampled the vertical structure of the surrounding environment, including one profile flown as a spiral located in a region completely free from any cloud, when possible.

A total of six process studies comprising individual research flights (RF) were conducted using this flight module design: (1) RF39 2020-09-29, (2) RF77 2021-06-02_L2, (3) RF80 2021-06-07_L2, (4) RF171 2022-06-10_L2, (5) RF173 2022-06-11_L2, and (6) RF176 2022-06-14 (Note: following the archiving convention L2 denotes the second flight of a given day). Here we will describe these respective flights as Cases 1-6. Cases 1-3 were flown from NASA Langley Research Center and Cases 4-6 were flown from Bermuda, where the project was based during June 2022. The flights originating from Bermuda benefitted from fewer airspace restrictions and shorter transit times to regions of interest resulting in longer loiter times for sampling. Consequently, Case 4 included a second cloud target that was fully sampled by the Falcon but partially sampled by the King Air without additional dropsondes. Case 5 comprised two modules that included both aircraft, but the initial



module was abbreviated by the Falcon because of rapid decay of the targeted cloud system. For instances where we wish to differentiate the two sections of these flights, they will be referred to as Case 4A/B and 5A/B, respectively.

## 2.2 King Air

A multi-wavelength airborne high spectral resolution lidar (HSRL-2; Hair et al., 2008; Burton et al., 2018) provided vertically resolved aerosol and cloud properties below the King Air altitude. The HSRL-2 generates simultaneous
measurements of particle backscatter coefficient and depolarization ratio at 355, 532, and 1064 nm as well as particle extinction coefficient at 355 and 532 nm, providing information about aerosol extensive and intensive properties. Aerosol type (marine, polluted marine, pure dust, dusty mix, smoke, fresh smoke, and urban) was derived using the depolarization ratio, spectral depolarization ratio, color ratio, and lidar ratio (Burton et al., 2012, Burton et al., 2014). Particle backscatter was used to diagnose the presence of liquid clouds and determine cloud top height at high vertical (~1.25 m) and horizontal
(~60 m) resolution.

The Research Scanning Polarimeter (RSP) is a passive downward facing polarimeter with nine spectral bands (410, 470, 550, 670, 865, 960, 1590, 1880, and 2260 nm) that scans along the direction of flight (nadir ± 55˚) providing retrievals of aerosol, cloud, and surface properties (Cairns et al., 2003). During conditions with a narrow viewing angle differential relative to the solar principal plane and absence of cirrus, the RSP was used to provide cloud top microphysical properties
including the drop size distribution for liquid clouds (Alexandrov et al., 2018).

The NCAR Airborne Vertical Atmospheric Profiling System was included on the King Air to acquire dropsonde observations, providing vertical profiles of temperature, humidity, pressure, and horizontal wind components. The dropsondes used here were the NCAR NRD41 "mini-sondes" (Vömel et al., 2021, Vömel et al., 2023).

A nadir-facing camera (Case 1: Garmin VIRB Ultra 30 and Cases 2-6: AXIS F-1005-E) was mounted underneath the
fuselage to provide continuous (1-2s frame rate) images of the cloud scene. The AXIS camera was fitted with different lenses that changed the field of view amongst cases resulting in a different footprint when viewed from 9 km. The camera images were also used to qualitatively diagnose the position of the aircraft during transects with respect to the target cloud system.

## 2.3 Falcon

Full details and specifications of the Falcon payload are described in Sorooshian et al. (2023) and here we provide a summary of the instruments. Water vapor was measured using an open path Diode Laser Hygrometer (Diskin et al., 2002) and temperature was obtained from measurements of total air temperature using a Rosemount 102 probe. Three-dimensional wind components were derived using a radome-mounted, inertially-corrected 5-hole gust probe (Thornhill et al., 2003; Barrick et al., 1996). Temperature, water vapor, and winds were acquired at 20 Hz. CO, $CO_2$ and $CH_4$ were measured using
a near-IR cavity ringdown spectrometer (Picarro G2401-m; DiGangi et al., 2021) and $O_3$ was measured by a dual-beam ultraviolet absorption sensor (2B Technologies, Model 205), all at ~2 s sampling interval.



Cloud droplet size distributions (DSD) were measured using a Fast Cloud Droplet Probe (FCDP; SPEC Inc.) and a 2D-S Probe (SPEC Inc.; Lawson et al., 2006) spanning 3-50 µm and 30-1500 µm drop diameter, respectively. The distributions from both instruments were merged onto a uniform logarithmic grid (smoothed to 20 bins per decade) and a weighted

average taken in the overlap region (30-50 µm), with weights smoothly transitioning from FCDP to 2D-S. The linear spaced size grid for the 2D-S, based on pixel occultation, results in low counting statistics for larger drops and is improved by re-binning. During Cases 1 and 2, there were some periods where 2D-S data were not acquired and precipitation DSD data from the Cloud Imaging Probe (CIP; Droplet Measurement Technologies) were substituted covering the size range 500-1500 µm, which represented the only range where image analysis could be conducted for the CIP. Thus, during these periods no

DSD data in the 50-500 µm range could be derived.

All clouds sampled by the Falcon in these process studies contained only liquid drops, as verified using particle imagery supplied by the 2D-S. The number concentration, $N_d$, liquid water mixing ratio, $q_L$, and precipitation rate were determined through integration of the DSD and terminal velocity data from Beard (1976), since in situ samples were verified as liquid drops. A notable caveat in relation to cloud phase exists for Case 6 where liquid-only supercooled drops at ~-8˚C were

observed near cloud top (~5.6 km) at the time of initial Falcon sampling. The Falcon progressed downward in altitude to warmer temperatures while continuing to observe liquid-only conditions, but the cloud system was concurrently observed by the King Air to grow to ~ 7 km (-15˚C). Hence the presence of ice in this system cannot be discounted across its lifecycle, despite no such indications from the Falcon observations. Cloud drop composition was directly measured through cloud water collection using an Axial Cyclone Cloudwater Collector (AC3; Crosbie et al., 2018), which usually resulted in one

sample per cloud leg. These samples were analyzed offline for major ions, pH, and elemental composition.

Dry aerosol particle size distributions were measured using the combination of a Laser Aerosol Spectrometer (LAS; TSI Model 3340, 100-3000 nm diameter) and Scanning Mobility Particle Sizer (SMPS; TSI Model 3085 DMA, TSI Model 3776 CPC, 3-100 nm diameter) stitched at 100 nm (Sorooshian et al., 2023). The LAS acquired size distributions at 1 s intervals while the SMPS performed 45 s scans. Here we report size distributions as averages over time periods that include several

SMPS scans. Condensation particle counters (CPC; TSI Model 3756 and 3772, respectively) provided ultrafine (>3 nm) and fine (>10 nm) total particle concentrations with an additional fine CPC downstream of a thermal denuder at 350˚C providing non-volatile particle concentration (>10 nm). A CCN counter (DMT Model 100; Roberts and Nenes, 2005) provided concentrations at 0.37% supersaturation. Two integrating nephelometers provided dried and humidified (< 1 µm) particle scattering at 450, 550 and 700 nm wavelengths (TSI Model 3563). Non-refractory aerosol mass concentrations (< 1 µm)

were measured using a high-resolution time-of-flight Aerosol Mass Spectrometer (AMS; Aerodyne Research Inc.).

The FCDP was also used for characterization of super-micrometer aerosol during sampling of clear air ($q_L$ < 0.001 g kg$^{-1}$, no precipitation, RH < 95%). This provided particle number and volume estimates (i.e., analogous to those described above for the LAS) at ambient conditions extending to larger sizes to aid characterization of coarse aerosol.

The Falcon was also equipped with a forward-facing camera and a downward-facing Heitronics KT-15 Infrared

Thermometer used to determine SST.





## 2.4 Auxiliary Datasets

### 2.4.1 MERRA-2

Instantaneous (3 hour) three-dimensional meteorological fields from the NASA Modern-Era Retrospective Analysis for Research and Applications Version 2 (MERRA-2; Gelaro et al., 2017; last access February 2, 2023) were used to provide
supporting synoptic scale data for the large-scale environment surrounding each case. Large scale winds, temperature, humidity, and geopotential height data are available on interpolated pressure levels at 25 hPa intervals (in the lower troposphere) and 0.625˚ x 0.5˚ grid spacing. Two-dimensional fields (surface pressure, sea level pressure and surface geopotential) are provided on a collocated grid and used to determine the lower boundary (e.g., for adjacent land masses) and extrapolate the 1000 hPa geopotential height when below the surface. Precipitable water (PW) was estimated by trapezoidal
integration of:

$$PW = \frac{1}{g} \int_0^{p_{sfc}} q_v \, dp \qquad (1)$$

and the horizontal column moisture flux (MF) similarly calculated using:

$$\boldsymbol{MF} = \frac{1}{g} \int_0^{p_{sfc}} \boldsymbol{U_h} q_v \, dp \qquad (2)$$

where g is the gravitational acceleration, p ($p_{sfc}$) is (surface) pressure, $q_v$ the water vapor mixing ratio, and $\boldsymbol{U_h}$ is the
horizontal vector wind. Contributions to the integral below 1000 hPa were included by using the values of the fields at 1000 hPa.

Airmass trajectories were derived from the MERRA-2 horizontal wind fields averaged over a vertical slab between 950 hPa and 800 hPa to reflect the dominant low-level horizontal motion. The airmass trajectory was solved numerically by integrating the slab wind forward and backwards in time using a linear interpolation of the wind field to the trajectory
location and time within the 3-hourly reanalysis outputs. This method of calculating trajectories intentionally ignores reanalysis vertical motion because the main purpose of the trajectories (in the backward direction) was specifically to assess surface source regions and history. Note that slab winds excluded the 975 and 1000 hPa levels to minimize sensitivity to reanalysis near-surface wind structure.

### 2.4.2 GOES-East Advanced Baseline Imager

Visible (0.6 µm) satellite imagery from the Advanced Baseline Imager, onboard the 16[th] Geostationary Operational Environmental Satellites (GOES-East) was accessed through the NASA Langley Satellite Cloud and Radiation Property Retrieval System (SatCORPS; last access January 18, 2023). The high resolution (~0.5-1 km pixel size) imagery was captured over the duration of each flight and relevant adjacent time periods at 20-minute intervals.

### 260   2.4.3 Gridded Sea Surface Temperature



Global daily Group for High Resolution Sea Surface Temperature (GHRSST) gridded data at 0.01˚ spatial resolution were acquired for each flight (GHRSST Level 4 MUR dataset; last access October 18, 2022). The GHRSST dataset combines night time multi-platform satellite-retrieved products with in situ buoy SST measurements (Chin et al., 2017).

## 3 Synoptic Environment

Backward trajectories (Figure 2a) implied a tropical marine airmass origin (over a time period of 8 days) across all cases originating from the central or western tropical Atlantic and conforming to the expected climatological circulation around the subtropical anticyclone. Based on proximity to the North American continent, the airmass history of Case 3 indicated the potential for contributions from continental pollution sources and indeed trajectories extracted for higher altitudes (above 850 hPa, Figure S1) indicated outflow from the eastern United States. Cases 1-3 were all located along the axis of the Gulf

Stream (Figure 2b), while the remaining cases were situated over more spatially uniform surface conditions near Bermuda. Data shown in Figure 2b relate to Case 6, but general SST spatial patterns were broadly consistent amongst the other cases (Figure S2), with the caveat that Case 1, which occurred in September, experienced regionally higher SST consistent with the seasonal cycle. The synoptic meteorological environment was broadly similar across cases and an example of the large scale pattern is shown for Case 6 (Figure 2c,d). The center of the subtropical anticyclone (as diagnosed by sea level

pressure) was located to the east and a quasi-stationary frontal boundary was located to the north, marking a region containing deep convection with enhanced PW and MF (Figure 2d). Comparisons with other cases can be found in the Supplement (Figure S3). During the majority of the summertime campaign this frontal boundary was a persistent feature, with its position, strength, and characteristics modulated by transient mid-latitude systems and was often anchored to surface features such as the Gulf Stream or the coastal region of the United States. Across cases, the relative position of these major

synoptic features remained broadly consistent with respect to the location of the aircraft sampling such that, for Cases 1-3 (located closer to the continent), the anticyclone was farther west with deep convection and frontal cloud generally found near the coast and onshore.

## 4 Mesoscale cloud organization and environment

### 4.1 Satellite Tracking

Each process study flight module was fixed to a visually selected cloud feature that was used by both aircraft as a reference and defined the center point of the sampling region. By design, satellite imagery taken near the midpoint time of each module indicated regions of enhanced cloudiness and visible cloud aggregation collocated with the sampling region (Figure 3), which was approximately bounded by the area spanned by the dropsondes (yellow dots). Most apparent in Cases 1 and 2 but also observed in Cases 3 and 5 was the prevalence of very small cumulus fields elsewhere in the cloud scene and in the

periphery of the enhanced cloud regions together with the emergence of cloud free zones often forming in the immediate





surroundings. The position of the Falcon spiral profile (orange cross) was usually in one of these clearings 20-40 km from the cluster centroid. Contours of SST indicate the relative position of the module to the axis of the Gulf Stream in Cases 1-3 and the comparatively homogeneous SST in Cases 4-6. In Cases 1 and 2, cloud sampling occurred above the ridge of maximum SST, while Case 3 sampled the cloud system as it crossed the sharp gradient on its northwest edge (0.2 K km$^{-1}$).

The cloud cluster anchoring each flight module was tracked using satellite imagery by calculating the maximum cross-correlation associated with sequential images for a region surrounding the cloud cluster (Nieman et al., 1997). Sequential images were analyzed forward and backwards in time to estimate the lifecycle of the feature and the displacements were fit using least squares regression to create a first order (i.e., linear) prediction of the cloud motion zonal and meridional velocity components over the observed lifetime (Table 1). Satellite animations of the cloud scene evolution viewed in the derived

(moving) reference frame of the cloud cluster are included as a video supplement.

The longevity of each trackable feature varied from less than two hours (the truncated Case 5A) to more than eight hours. Cases 1 and 2 exhibited the longest-lived features and the genesis and lysis were less conclusive because Case 2 was subsumed into deep convection and Case 1 likely shared the same fate but was first obscured by an over-running altocumulus deck approximately two hours after the aircraft sampling concluded. The lifecycle of the other cases was more

obvious because before and after there were either no clouds or limited/negligible aggregation. In Case 4 (Figure 3d) the image corresponds to the midpoint of the primary module (Case 4A), which encompassed most of the mature lifecycle of this cluster. The secondary module located just to the south (Case 4B) was part of the same mesoscale region and the location of this cloud cluster at the time of the image is shown (marked "x"). Figure 3e shows the image at the midpoint of Case 5B, which became the primary module due to the rapid decay of Case 5A and whose clouds no longer exist at the time

of the image.

## 4.2 Thermodynamic Profiles

Mean dropsonde vertical profiles of potential temperature (θ) (Figure 4) show similar vertical structure amongst cases, in line with expectations for the summertime lower and mid troposphere in this region. Using case-specific parcel properties

representative of the mixed layer, all contained moderate (marginal) convective available potential energy (CAPE) when implementing a pseudo-adiabatic (reversible) assumption and were associated with minimal convective inhibition (CIN) (Table 2), as is typical of tropical oceanic soundings (e.g., Betts, 1982, Xu and Emanuel, 1989). Although stable layers were present in some cases, notably Case 1 (Figure 4c), other cases (e.g., Case 5) showed little change in stability with altitude. From stability alone, there were no indications from the environment precluding deeper convection from developing and

indeed, for all cases, satellite imagery indicated deep convective cells within 200-400 km.

Mean water vapor mixing ratio ($q_v$) indicated more variability amongst cases particularly in the 2-5 km altitude range, which coincided with the region of maximum saturation deficit (e.g., compared to the reference wet adiabat) and PW varied from 35-48 mm (Table 2). Overall, there was not a clear qualitative relationship between the details of the moisture profile and the





size and extent of the cloud aggregation, as visualized by satellite (Figure 3). For example, Case 5 exhibited the smallest

cloud features with the least overall cloud coverage (Table 2) yet the environment of Case 5B had the highest PW.

## 4.3 Winds

Wind profiles are shown in Figure 5 as hodographs to visualize the effects of vertical shear of the horizontal wind and the winds relative to cluster motion. Relative winds can be visualized by shifting the origin to the cluster motion (square

markers) and cloud layer shear can be represented by the vector from cloud base (down triangle) to top (up triangle). All cases indicate that cloud motion was close to the cloud base wind vector (within 1.3 m s$^{-1}$ relative vector magnitude), while the magnitude of the wind shear between cloud base and top varied from 1.6 m s$^{-1}$ (Case 4) to 7.6 m s$^{-1}$ (Case 2). In Cases 1, 2, 5 and 6, the cluster motion was aligned (within 15˚) with the major axis of cloud organization (indicated as a dashed line in Figure 5), but only in Case 6 was there an absence of any directional shear such that the cluster motion, the shear vector,

and the major axis were all aligned. Cases 1 and 5 exhibited several near-parallel linear cloud features ("cloud streets") and shared a similar near-perpendicular shear vector across the depth of the cloud and it is notable that pronounced linear organization occurred in Case 5 despite both weak mean flow and shear. Shear was almost not existent in Case 4 along with only marginal directional organization of cloud features within the module, although there was evidence of cloud banding at a larger scale (~1000 km) along a north-south axis east of the module. Case 3 was the only example where the cloud formed

a pronounced linear (~ 40 km) feature that was oriented approximately perpendicular to the cluster motion and the shear vector, although satellite imagery also indicated that this was part of a larger (~ 200 km) organizational pattern of interconnected rings extending along the Gulf Stream axis to the northeast.

## 5 Cloud Properties

### 5.1 Cloud height distributions

Frequency distributions of the heights of cloud tops detected by HSRL-2 during the King Air module (Figure 6) indicated that in four of the cases (Cases 1-3, 5) a pronounced peak in frequency occurred below 1 km with a near-monotonic decrease with altitude in the 1-2 km thereafter. Across these four cases, the modal altitudes were 100-200 m above the altitude of the lowest cloud base determined from the low-flying Falcon (Table 2) and reflected the relatively high occurrence of very small cumulus anchored atop the marine sub-cloud mixed layer that prevail in peripheral regions surrounding cloud clusters,

despite the apparent emergence of clearings on satellite imagery. Mixed layer depths (Table 2) were determined from the altitude of the first maximum in relative humidity of the mean dropsonde profile, which also corresponded to the inflection altitude where mean θ ($q_v$) substantially increased (decreased) (Figure 4) and were found to be within 50 m of the lowest cloud base. Across these four cases, the lowest lifting condensation level (LCL) determined from the mean dropsonde data was found 70-140 m above the lowest cloud base and attributable to variability in temperature and humidity, while increases





in mixed layer depth and the cloud base correlate with increases in the cloud top modal altitude at a degree slightly higher than proportional (1.2) indicating a minor thickening of the small cloud mode with increasing sub-cloud depth.

In stark contrast, the dominant cloud top modes for Cases 4 and 6 were at 2.9 and 2.5 km, respectively. The vertical distribution of clouds was fundamentally different in these cases and pronounced local maxima in the frequency of cloud tops were observed within multiple altitude ranges, while cloud tops observed below 1 km were less frequent, most notably

for Case 4. Undoubtedly, some low-lying clouds may be obscured by laterally spreading cloud aloft, and these cases also generally sustained fewer small cumulus in the surrounding environment, as indicated in Figure 3 and as confirmed by in-flight camera imagery. The observations indicate that regions of enhanced stability do not exert an obvious controlling influence on the distribution of cloud tops, except Case 4. Regions of the column with static stability exceeding 6 K km$^{-1}$ (shaded in Figure 6) would tend to inhibit further cloud growth, creating the expectation for more cloud tops to be found

within those regions. However, local maxima in cloud top frequency are almost equally located above, below, and within stable layers, while some stable layers result in no apparent influence on the vertical distribution entirely.

The 0˚C altitude (marked in Figure 6) has been attributed to a stability enhancement (e.g., Posselt et al., 2008) but in these six cases there was not a conclusive association between stability and the region close to the 0˚C level; however, Cases 1 and 6 (and marginally Case 4) indicated some partiality for cloud tops at that level. It is worth noting that the clouds detected

above 4 km in Case 1 were mostly associated with altocumulus that was not coupled to the convection below, except in one region (confirmed via camera imagery) that was associated with a short-lived deeper convective cell, offset from the Falcon's sampling region. In contrast with apparent influence seen in Case 6, Case 2 did not show any relationship between clouds, stability, and the 0˚C level, while no clouds were observed to reach that level in Cases 3 and 5. The cloud tops observed above 4/3/6 km in Case 2/5/6 were determined to be associated with emergent growth of convective turrets that

occurred after the start of the Falcon module and so those altitudes were not sampled in situ. Less cloud was observed at all altitudes during Case 5A compared to 5B in agreement with the initially small size of this cluster at the onset of the module and the subsequent decay. Cloud height distributions can also be visualized using the HSRL-2 backscatter curtain plots, which are included in the Supplement (Figure S4).

**5.2 Vertical Velocity**

Figure 7a shows statistics of the vertical distribution of vertical velocity, w, measured in situ for each case. Each level leg was truncated to remove maneuvers and high-pass filtered (50 s FFT filter) to remove low-frequency biases/drift and any residual airframe dynamical effects. A $q_L$-weighted mean was calculated for w, which represents a characteristic velocity for the convective transport of condensed water (here $w_L$ is used to specifically differentiate this quantity). The spread of w,

indicated by the 10-90% range, quantifies the relative magnitude of cloudy updrafts and downdrafts (filtered for cloud, $q_L > 0.02$ g kg$^{-1}$).

Across the cases, maximum updrafts varied between 1.1 m s$^{-1}$ (Case 4A) and 9.9 m s$^{-1}$ (Case 3), while maximum downdraft magnitudes varied between 1.1 m s$^{-1}$ and 6 m s$^{-1}$ (the same cases, respectively). Updraft and downdraft extrema were all





located within 850 m of the highest in-cloud transect, but because cloud top was dynamically evolving it was challenging to classify this level as a fraction of cloud depth. Transects that encountered stronger updrafts also contained stronger downdrafts, resulting in a correlation coefficient, between updraft and downdraft velocity, of -0.77. The mean updraft velocity exceeded the mean downdraft by 17% in magnitude.

Positive $w_L$ occurred in 82% of cloud transects and corresponds to a conventional expectation of strong upward flux of cloud water in the convective core with subsiding motion in diluted (i.e., lower $q_L$) peripheral regions and the immediate cloud-free environment. Negative $w_L$ occurred predominantly near cloud top in a subset of cases and is believed to be an entirely transient characteristic of individual convective turrets because the source of cloud water is condensation in updrafts. Transient negative $w_L$ may occur when cloud parcels descend after having overshot neutral buoyancy or because of negative buoyancy generated through homogenization of entrained air. While a descending cloud top interface does not necessarily require negative $w_L$, transient negative $w_L$ is associated with a cyclical collapse of the cloud top and this was observed visually during flight, most prominently in Case 3. Here, the upper extent of the cloud rapidly evaporated and descended approximately 1 km in 5 min, as diagnosed by forward camera imagery, thus implying a cloud top recession velocity of $\sim$-3.3 ms$^{-1}$, compared to the observed $w_L$ of -1.6 ms$^{-1}$ measured during the uppermost transect. Case 5A contained the highest occurrence of negative $w_L$ and within 15-20 min there was no visual evidence of the cloud system, which was confirmed by visible satellite imagery. These cases are too limited in number to provide a statistical description of the behavior of an ensemble of cloudy thermals within a typical system but the occurrence of transient negative $w_L$ here may be amplified as a consequence of the flight strategy. Emergent convective turrets that were used to anchor the sampling were identified with a lead time of several minutes, so there was a greater chance (than typical) that a selected turret would subsequently be in decline by the time sampling of the cloud top region was underway. Evidence of thermal bubbles exists in the vertical structure of $w_L$ particularly for Case 3 and the upper half of Case 6. Regions where $w_L$ was close to zero also tended to occur in parallel with local reductions in the strength of updrafts and downdrafts at that level and are attributed to wake regions detrained from active thermals. While residual turbulence remained in these regions, parcels may be closer to neutral buoyancy representing the more aged regions between rising bubbles or regions where cloud had spread laterally. Beyond the scale of single transects and individual thermals, Case 4A (which was sampled immediately after deeper convection had ceased) exemplifies this at the cloud system scale with minimal $w_L$ and weaker updrafts and downdrafts, in agreement with the observed lack of surface-coupled convection and a more stratiform appearance.

### 5.3 Liquid Water Content

Vertical gradients of $q_L$ capture the rate of condensation mediated by the effects of dilution and evaporation from entrainment alongside losses due to precipitation. An adiabatic $q_L$ from a parcel released at the observed lowest cloud base for each case (included in Figure 7b) represents an estimate of the $q_L$ resulting from condensation alone, generally providing an upper bound. Near cloud base, $q_L$ increased with altitude with some cloudy parcels initially approximating the adiabatic parcel. With altitude, the envelope of $q_L$ (as quantified by the upper 90% bound) generally increased at a slower rate (Cases





1, 4B, 5), exhibited little trend (Cases 3, 6), or reverted to a decrease (Case 2). The highest $q_L$ was observed in Case 3 (2.7 g kg$^{-1}$) despite the fact that the environmental humidity was lowest (Figure 4b). For some upper cloud regions, $q_L$ tended to

fluctuate between adjacent sampling levels and this was most notable in Case 6, perhaps accentuated by the cloud system depth and the high number of cloud transects. This pattern is indicative of snapshots through an ensemble of transient convective elements (e.g., Morrison et al., 2020), where a chain of several rising thermals occurs sequentially, with each evolving in time, such that the aircraft transects reflect different stages of their lifecycle, proximity to their cores, and energetic characteristics.   In these dynamic environments, the aircraft measurements cannot singularly isolate vertical

variability but rather incorporate the temporal evolution of individual thermals and the cloud system at large. Despite uncertainty in segregating time evolution and spatial scales of variability, the aircraft snapshot confirms an intermittent structure for emergent cores rather than a stationary plume, while at lower altitudes the data are more reflective of the larger ensemble of contributing thermals at each level.

**5.4 Cloud droplet number concentration**

The overall trend was for droplet number concentration ($N_d$) to decrease with increasing altitude, except Case 5 where a minor increase was observed (Figure 7c).  In Cases 1 and 3, the trend in $N_d$ was punctuated by locally anomalous high concentrations at 2.1 and 3.0 km, respectively, corresponding to local enhancements in the profile of $q_L$ and were attributable to fresh, energetic, convective bubbles.  In absence of these singular outliers, Case 1-3 (and Case 6 below 2.5 km) indicated

near monotonic decreases at a rate of between 27-52% km$^{-1}$. Comparison of the mature/decaying (4A) and active/developing (4B) systems of Case 4 showed very close agreement in average $N_d$ across commonly sampled altitudes, indicating robustness in the structure of the profile and suggesting temporal evolution of horizontal mean $N_d$ may not be substantial over the system lifecycle. Cases 4B and 6 exhibited similar vertical patterns with decreasing $N_d$ found below 3 km with subsequent increases to a secondary maximum aloft at 4.3 and 4.6 km, respectively.

The source of $N_d$ at cloud base is the activation of aerosol particles that serve as cloud condensation nuclei (CCN). The number concentration of particles with diameters exceeding 100 nm ($N_{a,100}$) was used as a proxy for the availability of CCN, qualitatively captured the variability in $N_d$ amongst the cases.  The $N_d$ for Case 5A (decaying) was 45% lower than Case 5B (active) but coincided with a 22% decrease in the below-cloud $N_{a,100}$.  The fact that the change in $N_d$ was proportionally greater than the change in aerosol may be explained partly by weaker cloud forming convective updrafts in Case 5A, even

though the sub-cloud turbulence was found to be similar (Table 2). Crucially, a greater desiccation by entrainment because of its smaller size and lack of fresh convection was likely controlling the statistics of $N_d$ observed in Case 5A.  In an attempt to separate initial activation (a topic that will be discussed further in Section 7) from subsequent in-cloud controls on $N_d$ variability, a reference concentration ($N_{d0}$) was calculated from air parcels near cloud base that indicated recent droplet activation (Conant et al., 2004).  Here, $N_{d0}$ (Table 2) was computed as the mean of data points collected in the lowest two

cloud transects, limited to updrafts free from precipitation, and with $q_L$ within 80% of the reference adiabatic parcel



representing a best estimate of the initial cloudy state. The $q_L$ criterion was relaxed to the maximum observed adiabatic ratio when no data above 80% occurred.  The lowest $N_{d0}$ was found for Case 1 (144 mg$^{-1}$) and the highest for Case 3 (508 mg$^{-1}$). $N_d$ data were normalized by $N_{d0}$ and then compared to the ratio of $q_L$ and the adiabatic water content (Figure 8), representing a comparison between the observations and an idealized adiabatic parcel. Phrased another way, these normalizations are the

fraction of drop number and mass concentration retained, compared to an undilute parcel ascent.  Mean properties from each transect (Figure 8a) indicate that reductions in $q_L$ result in proportionally smaller changes in $N_d$ such that 96% of the transects lie above the 1:1 line.  For clarification, a 1:1 relationship would indicate that number and mass are equally affected by dilution and evaporative losses, suggesting either extreme inhomogeneous mixing or partial volumes of cloudy and clear air at the scale of the measurement (~100 m).  There is no strong divergence or separation in the behavior between cases

indicating some degree of universality: the combined data exhibit a positive correlation coefficient (R=0.73), and a total least squares linear regression indicates a slope of 1.89.  The joint frequency of 1 s cloudy data (rather than transect means) across all cases (Figure 8b) confirms the same enhancement of $N_d$ over a 1:1 relationship and, for the lower range of $q_L$ that comprises the majority of the data, suggests that a linear model is appropriate, noting that logarithmically spaced bins were used to better reflect the distribution of data points.  At higher $q_L$, the use of 1s data shows that the distribution asymptotes

towards $N_{d0}$, in line with expectations for data to coalesce around (1,1), by design. Although data points that are close to adiabatic represent a small set of the total observations, separation by vertical velocity (Figure 8c) indicates that this region of the joint histogram is more influenced by updrafts.

All else held constant, droplet collisions would reduce $N_d$ without changing $q_L$, while loss of $q_L$ by accretion of falling precipitation would have a near equivalent impact on $N_d$, notwithstanding a strong size dependence of collection efficiency.

Therefore, the results shown in Figure 8 indicate that collisions cannot be dominant in shaping the budget and vertical distribution of $N_d$, particularly for updrafts.

## 5.5 Drop Size Distributions

Anomaly drop size distributions (DSD) were derived from the mean DSD from each cloud leg, normalized by leg mean $N_d$,

and then compared as a ratio to the normalized mean of all cloud legs (Figure 9).  The resulting anomaly DSD vertical profile represents relative enhancements or reductions in the DSD shape compared to the reference DSD shape, without conflating changes in $N_d$.  The advantage of this approach is that anomalies can be assessed independent of the drastic change in number concentration that occurs over the entire DSD (i.e., both cloud and rain water modes).  At each level, the DSD exhibits the combined influence of net condensation, entrainment and subsequent mixing of environmental air,

secondary droplet activation, collisions amongst cloud droplets and the removal of drops by falling precipitation.  A monodisperse reference drop diameter was calculated using $N_{d0}$ and $q_{L,ad}$ to represent idealized behaviour of a reference adiabatic parcel and is overlaid on Figure 9.

We observe three major modes that vary in their degree of significance amongst cases: (i) a condensation/evaporation mode that resembles a "J" shape and closely, but not identically, aligns with the reference monodisperse diameter and could



represent upward or downward motion, (ii) a precipitation growth mode that when coupled with the upper section of the "J"
forms an inverted "V" and reflects rain drop growth caused by accretion, and (iii) a secondary activation mode that occurs to
the left of the "J" and may occur at multiple altitudes.

Positive anomalies associated with mode (i) often track close to the monodisperse adiabatic diameter even with the sub-
adiabatic profile of $q_L$ (Figure 7b) and more importantly, despite the proportional enhancement of $N_d$ (Figure 8). Extreme

inhomogeneous mixing of entrained environmental air would tend to affect $N_d$ and $q_L$ equally by completely evaporating a
subset of drops while the remaining drops retain their size (Jensen and Baker, 1989). Conversely, homogeneous mixing
reduces the size of all drops and, in isolation, could offer a partial explanation for the behaviour shown in Figure 8 where
drops lose cloud water or do not grow as fast (reduced $q_L/q_{L,ad}$), but with a lesser impact on $N_d$ (less reduced $N_d/N_{d0}$).
However, the near adiabatic growth of mode (i) (Figure 9) would contradict the assertion; instead indicating that a subset of

droplets was shielded from the influence of entrainment. The majority of 1 s observations show that $q_L$ remains distinctly
sub-adiabatic, even at the 90% level (Figure 7b), suggesting that any regions of undiluted growth manifest predominantly at
scales <100 m.

Mode (i) is often accompanied by additional smaller drops (i.e., explaining the enhanced $N_d$ behaviour of Figure 8) that
sometimes are concentrated at specific sizes (i.e., as mode iii). In addition, the reference normalized DSD in Figure 9 shows

pronounced multimodal characteristics that would not be as distinct if it were only representing the averaged growth with
altitude of a single broadened mode. While some contributions to mode (iii) may result from mixing that is effectively
homogeneous at the smallest scales but inhomogeneous at larger scales (but still <100 m), the magnitudes of the
enhancements in mode (iii) are suggestive of episodic, distinct, secondary droplet activation (i.e., events that take place
above the lowest cloud base).

The broadening of the DSD with altitude, implied by Figure 9 through the emergence of modes (i) and (iii), was investigated
quantitatively using the relative dispersion, $\epsilon$, which relates the standard deviation of drop sizes comprising the DSD to the
mean size (Tas et al., 2015). As a number-based measure of DSD broadening, the magnitude of $\epsilon$ is insensitive to the tail of
DSD and therefore not directly impacted by processes relating to precipitation (i.e., the role of mode (ii)). Amongst cases, $\epsilon$
exhibited similar values (Figure 10) showing a consistent increase with altitude (0.075 km$^{-1}$). Across the data, the average $\epsilon$

was 0.47±0.12, which is higher than reported for cumulus over land (Tas et al., 2015), potentially because their cases were
shallower and more polluted. Furthermore, the updraft dynamics of daytime cloud-forming thermals over land may result in
fundamentally different entrainment-microphysics interactions compared to these marine cases. In summary, significant
DSD broadening is attributed to entrainment processes; specifically, the combination of inhomogeneous or incomplete
mixing of rising parcels together with activation of additional droplets within the time that mixing is taking place.

In all cases there was evidence of precipitation initiation/formation near cloud top, attributed to an active
collision/coalescence process causing growth of the anomaly DSD well beyond the adiabatic diameter (Figure 9). However,
the emergence of subsequent precipitation growth (mode ii) was most prominent in Cases 2, 4B, and 6, though each case
exhibited a marked decrease in the significance of this mode at lower levels. Part of the decrease of the rain mode was





attributed to the higher fractional contribution of non-precipitating clouds that did not extend beyond lower altitudes (Figure
6), but it is notable that some cases (3, 4A) exhibited distinct breaks in the precipitation growth mode and may represent the
temporal and spatial intermittency of convective transport.

A hypothesis for the rapid decrease in the precipitation mode at lower levels (seen most prominently in Case 3) is that time
needed to produce precipitation and enact sufficient growth is in direct competition with the buoyancy "clock" of cloudy
volumes, which are progressively succumbing to the effects of continual entrainment and turbulent mixing. Older cloudy
volumes that contain nascent precipitation carry a risk that fresh convective bubbles, that carry a buoyancy premium, rise
through their midst, expel them laterally, and promote their evaporation into the environment rather than continuing to
accrete cloud water. Figure 11 shows four sequential transects from Case 3 between 2-3 km altitude and covering the region
where the rain mode appears active in Figure 9. The Falcon position has been projected onto a cloud centric coordinate
system using the fitted cloud motion (Table 1) to adjust for drift and rotated such that the x-axis corresponds to the direction
of cloud motion. In each leg, a dominant updraft was collocated with a region of high $q_L$ defining the convective core with
adjacent cloudy downdrafts. Rainwater fraction (RWF) was calculated as the fraction of $q_L$ contributed from drop sizes
exceeding 100 µm indicating that while the core was rain free (very low RWF), the downdraft regions at the edge of cloud
featured high RWF. This was most prominent for transect #4 at 2.7 km altitude, where high RWF downdrafts were observed
on both the entry and exit from cloud in regions with moderate $q_L$, such that RWF enhancement in downdrafts cannot be
explained by evaporation of small drops alone.

A further aspect of the measurements of droplet microphysics was the availability of remote sensing retrievals to provide
additional context as well as performance evaluation. Detailed analysis of the performance of combined HSRL-RSP
microphysical retrievals is the topic of further study, and these cases provide unique datasets for that effort. We limit this
evaluation to the retrievals of effective radius ($r_{eff}$) from RSP (note the use of radius here, by convention, while diameter is
used everywhere else), which is provided at ~1.2 Hz and was combined with the median HSRL-2 cloud height within each
period. Statistics were determined for each case by separating the HSRL-2 cloud heights into eight equally sized subsets
from which the mean and 10-90% range of RSP $r_{eff}$ were calculated (Figure 12) and compared to the same statistics for each
cloudy transect sampled in situ by the Falcon. RSP tended to underestimate the effective radius profile in cases where there
was a dominant rain mode; this was clearly captured for Case 6 where there was a closer agreement with in situ data if rain
water contributions to $r_{eff}$ were omitted. There are two contributing aspects: (i) a lack of sensitivity to rain-sized drops by
RSP (Alexandrov et al., 2018) that can low bias the $r_{eff}$ for these cloud systems when RWF is high near cloud top and, (ii) the
lack of an optical signature, at RSP wavelengths, of the deep interior of the cloud where most rain-producing drop
interactions occur. Unlike the Falcon transects, which provide a statistical representation at each level (notwithstanding
biases introduced from transient cloud behaviour), the RSP statistics reflect the microphysics of the outer "crust" of the
cloud cluster.

**5.6 Sub-cloud precipitation**



Rain rates were determined during in situ sampling legs carried out below cloud base to estimate the significance of surface precipitation. In the absence of radar data, it is challenging to place the under-sampled aircraft data into a statistical context, therefore the reported precipitation reflects only a snapshot and may contain biases associated with the flight strategy. The upper size limit of the 2D-S (1.5 mm) also under-sizes the contribution from large rain drops, which may cause a low bias in calculated rain rates. Considering these caveats, evaluation of sub-cloud precipitation was more focused on the relative differences between cases than assessing the broader relevance of rain rate absolute values.

Maximum (90%) and mean rain intensities (Table 2) were calculated from DSD data for each sub-cloud altitude using a threshold precipitation rate exceeding 0.01 mm hr$^{-1}$ to define rainy regions. The spatial distribution of precipitation tended to be highly concentrated in visually identifiable rain shafts and the flight line was adjusted to fly through their (visual) center, where possible. In cases where multiple sub-cloud altitudes were flown, the leg with the highest rain coverage and rain intensity was retained. No sub-cloud precipitation was encountered during Case 1 and 5A and in Cases 5B and 3, the rain was concentrated in a single narrow region less than 1 km in horizontal extent. To the extent that was possible, camera imagery was used to confirm that no major region of precipitation was missed simply by the choice of flight track. Case 5B was unusual because the maximum rain intensity was comparatively high (3.87 mm hr$^{-1}$) but it was limited to a very narrow region (0.6 km) in an otherwise completely non-precipitating cloud line. Conclusions drawn from comparing the maximum or mean rain intensity were similar and, as a singular metric for identifying the significance of precipitation in each case, the mean rain intensity did not capture the extent of the rainy region and therefore would over emphasize precipitation in Cases 3 and 5. Conversely, the transect mean (not included) was heavily biased to the sampling details of each particular case (e.g. the time spent sampling the clear-sky region). With a desire to derive a characteristic rain rate comparable amongst cases, the fractional rain coverage was estimated as the ratio between (linear distance) rain coverage below cloud and the maximum horizontal linear extent of Falcon cloud sampling aloft at any altitude. The product of the fractional rain coverage and mean rain intensity provided a cluster rain rate (Table 2). While an area fraction would be a more desirable quantity by which to derive this measure, the aircraft sampling tended to align with a principal cloud axis and therefore provided limited information by which to assess a second spatial dimension, and any assumptions would need to be case specific. Using the derived cluster rain rate, cases ranged from non-precipitating (1, 5A) to a maximum for Case 4, which interestingly revealed similar rates for 4A (0.45 mm hr$^{-1}$) and 4B (0.52 mm hr$^{-1}$) despite their differing convective characteristics, maturity, and peak rain intensities.

## 6 Composition

### 6.1 Trace Gases

In situ bulk statistics (Table 2) and vertical profiles (Figure 13) of CO, $CO_2$, $CH_4$ and $O_3$ were derived for each case. Vertical profiles were separated into data collected during cloud penetrations (in-cloud), adjacent cloud free regions sampled between each cloud level (near field), legs below cloud base (sub-cloud) and the spiral profile (far field). Differences





between the near and far field data reflect the influence of the cloud system on the vertical structure of the environment in combination with pre-existing horizontal gradients. Differences between the in-cloud and near field data are caused by convective transport. For CO, the statistical significance of profile differences was reduced because the magnitudes of the CO variations were proportionally smaller compared to the instrument precision (5 $ppb_v$, 0.4 Hz).

Near and far field profiles were generally similar across cases except for Case 3. Near field $CH_4$ and $O_3$ most closely tracked
trends associated with the far field vertical structure but tended to filter smaller scale features, perhaps indicating the influence of convective mixing on the near field environment. This was generally more noticeable in the case of $CO_2$ where far field overall vertical gradients were less apparent. In Case 3, the far field profile was characterized by a significant vertical gradient at 1.8-2.0 km where CO, $CH_4$, and $O_3$ rapidly increased (45, 50, 35 $ppb_v$, respectively), while $CO_2$ decreased (4 $ppm_v$). Above this altitude, $O_3$ continued to increase while the trend in other species reversed. This pattern was
a result of an airmass of marine origin undercutting a continental airmass aloft that had the signatures of anthropogenic influence coupled with a reduced $CO_2$ background caused by summertime biogenic uptake. The location of Case 3 on the gradient between polluted and background airmasses meant that the near field and in cloud concentrations were affected by both vertical and horizontal mixing.

In-cloud concentrations generally exhibited a smaller dynamic range, in line with expectations that cumulus serve to
transport sub-cloud air upwards and vertically homogenize near field environmental air through entrainment and hence their concentrations reflected a weighted average of sub-cloud and near field properties. In most cases there were many combinations of "weights" that could explain the in-cloud concentrations, but in some cases it is possible to do so while restricting contributions to the same level and below. Such scenarios are necessary for the archetypal rising entraining plume model of cumulus, such that in-cloud concentrations lag vertical gradients in the environment. However, there were
identifiable cases where the in-cloud concentrations led the environmental gradient, such as $O_3$ and $CH_4$ in Case 3 and $CO_2$ in Case 2 and (marginally) Case 6. These cases required air from higher altitudes to explain the in-cloud concentrations and hence suggests that simplified descriptions of lateral entrainment in shallow cumulus (e.g., de Rooy et al., 2013) should also account for (i) laterally entraining cloudy downdrafts, which were ubiquitous in all these cases (Figure 7), and (ii) the temporal characteristic of thermal bubbles. Further development and testing of theory and models for cumulus entrainment
and detrainment is not the focus here, but these case studies provide comprehensive datasets on which to base such an effort.

### 6.2 Aerosols

Aerosol optical property typing provided by HSRL-2 (Burton et al., 2014) indicated the fractional contribution of aerosol types (classified as marine, polluted marine, urban, smoke, fresh smoke, dust, dusty mix) by altitude during the entire King
Air module (Figure 14a). The "unclassified" type represents mixes or a lack of typable signatures, while the remaining absent fraction was not typable because of a lack of aerosol scattering, obscuration by cloud, or missing lidar data. A primary signature of marine aerosols was observed for Cases 1-3 with a secondary contribution from continental sources, typed mostly as smoke, influencing layers between 1-2.5 km (particularly in Case 3). Cases 5 and 6 indicated a dominant





dust layer, with the influence of marine aerosols taking a secondary role confined to the lowest 500 m. The dust was
assumed to be of Saharan origin based on back trajectories and persisted during other ACTIVATE flights during this time
period (i.e., flights that were not process studies). The smoke classification, and particularly the identification of fresh
smoke, during Case 4 was inconsistent and difficult to reconcile because of the absence of candidate sources and other
signatures that are expected for smoke such as elevated sub-micrometer organic aerosol mass (Figure 14c) and CO (Figure
13a). The back trajectory and the consistent synoptic pattern across the five days that included Cases 4-6 would create an
expectation for Case 4 to share the dust classification of Cases 5 and 6, and indeed aerosols were typed as dusty mix during
part of the statistical survey flight in the morning of June 10 (not shown). Both dust and smoke are depolarizing, and it is
possible that a misclassification could result from a dusty mix that is more aged, perhaps more coated with secondary
aerosol, and contains a higher (optical) influence from accumulation mode particles (of any source) that decreases the
particle depolarization and increases the lidar ratio.

Aerosol extinction from HSRL was mostly confined to the lowest 2 km of the atmosphere (Figure 14b), with Cases 1-3
indicating a more prominent enhancement in the lowest 1 km. With a deeper marine mixed layer than other cases, Case 2
exhibited the characteristic signature of aerosol hygroscopic growth leading to a defined maximum in extinction near the top
of the mixed layer; a pattern also seen in Case 1 and 3 but associated with a layer that was shallower. In situ sub-micrometer
dry extinction was computed from the sum of particle absorption and nephelometer total scattering measured at two relative
humidity (RH) levels and was scaled to RH=20 % using a gamma hygroscopic growth model. The contribution from sub-
micrometer aerosol water was estimated by determining the sub-micrometer extinction at the ambient RH and subtracting the
dry component. The super-micrometer extinction was estimated from particle area derived from the FCDP particle size
distribution (assumed to represent ambient RH) and an extinction efficiency calculated from Mie Theory assuming a
refractive index for water (1.33±0i). Overall, the in situ components of the ambient extinction capture the shape of the HSRL
profile but the HSRL profile is generally enhanced in overall magnitude by between 21% (Case 6) and 77% (Case 3). In
Cases 1-3, aerosol water contributed between 36% and 47% of the dry extinction to the total sub-micrometer budget within
the lowest 1 km, which spanned the mixed layer and the lowest part of the layer occupied by cumulus. It is expected that
water would contribute a similar, or larger, component of the super-micrometer extinction for these cases, based on the
expectation of sea spray dominance. Both in situ and HSRL data were screened to remove clouds, but both respective
methods have unavoidable differences. In addition, the relationship between aerosol water and humidity between 90% and
saturation is strongly convex meaning that contributions from air parcels near saturation have a highly disproportionate role
and could easily explain the differences seen at low altitude in Cases 1-3. Aerosol water contributed minimally to sub-
micrometer extinction in Cases 4-6, consistent with the expectation of dust. The super-micrometer extinction estimates are
likely to be more uncertain in these cases because (a) the sizing obtained from the FCDP was derived from an assumption of
spherical water droplets, and (b) particles between 1 µm dry diameter and the smallest observable FCDP diameter at ambient
humidity would be under-reported.





Non-refractory sub-micrometer particle composition (Figure 14c) was dominated by sulfate in all cases, except Case 3 where some layers were more organic dominant in alignment with a greater influence of continental pollution. While the organic contribution for other cases was too small to further interrogate the characteristics, the Case 3 organic aerosol indicated a lower degree of oxidation, as quantified by the $f_{44}$ AMS mass fraction, in the region below 1 km (0.10) compared to the 1.5-2.5 km range (0.15). The mass ratio of ammonium to sulfate did not have an observable vertical change and ranged from 0.12-0.27, with Case 1 and 2 less neutralized than ammonium bisulfate, Case 3 approximately equal to ammonium bisulfate and Cases 4-6 lying between bisulfate and fully neutralized ammonium sulfate.

## 6.3 Cloud drop composition

Cloud water composition (Figure 14d) was strongly influenced by sea salt, assumed to be a direct consequence of cumulus clouds lofting air from the marine mixed layer below cloud base and in broad agreement with other recent cloud water measurements of vertically developed cumulus (Crosbie et al., 2022; Stahl et al., 2021). The largest component of cloud water ionic mass can be attributed to sea salt in Cases 1-3 at all altitudes (89-94%, 49-67% and 51-66%, respectively) and is also in support of the increased prevalence of HSRL marine aerosol types below cloud (Figure 14a). The emergence of nitrate in the cloud water composition of Cases 2 and 3 (24-31% and 12-27%, respectively) occurs with an increase in the polluted marine designation by HSRL, perhaps confirming additional influence of anthropogenic sources compared with Case 1. Non sea salt sulfate (4-8%) and organic ions (1-9%) were proportionally enhanced during Case 3, also supporting continental pollution influence. Non-sea salt calcium, a common tracer for dust, was more enhanced in Cases 4-6 (0.9-9%) than 1-3 (0-2%), and, interestingly, this was also accompanied by an increase in non-sea salt potassium in the middle and upper regions particularly for Case 6 and contributed up to 20% by mass. Dust contains a greater fraction of insoluble material which does not contribute to the analysis of cloud water ions. Case 4 and Case 6 also exhibit larger vertical changes in the cloud water composition and that is notable because these cases were deepest and most affected by precipitation, potentially providing a mechanism for increased vertical stratification in drop composition. Enhanced sea salt in energetic cumulus tops has been observed elsewhere (e.g., Crosbie et al., 2022) and featured here mainly in Cases 2, 4B and 6. Notably though, clean regions at higher altitudes and the influence of precipitation removal on solutes may partly explain enhanced variability in relative composition, as small perturbations can generate large influences. Nonetheless, consistency across several sequential data points adds to the robustness of the observed trends.

## 7 CCN Activation and aerosol microphysics

### 7.1 Sub-cloud Aerosol

In situ aerosol particle size distributions were compiled for the sub-cloud environment in each case and normalized to unit integral (Figure 15a), showing the merged contributions from the SMPS and LAS. Across all cases the distributions reveal a clear multimodal structure with dominant contributions from Aitken and accumulation modes, separated by well-defined



minima (Hoppel et al., 1986), except Case 3 where the Aitken mode is broader and slightly larger. The less pronounced
Hoppel minimum found in Case 3 is aligned with expectations for enhanced influence from continental pollution, which may
have undergone proportionally less cloud processing.

Contributions to each mode are fit to log-normal distributions and parameters are provided in Table 2. Across cases, the
Aitken mode varied from 43 to 64 nm, while the accumulation mode varied from 180 to 230 nm. An activation diameter
was estimated as the size above which the integrated number concentration would equal the estimated $N_0$ (Table 2) and
ranges from 52 to 74 nm. CCN counter data at 0.37% supersaturation are included in Table 2, but closure of these data to
the particle size distributions would require unrealistically high particle hygroscopicity for Cases 4-6 based on equilibrium
drop activation calculations (not shown). In lieu of a detailed droplet closure analysis, the LAS number concentration (>100
nm), $N_{100}$, is quite effective at predicting $N_{d0}$ using a constant scaling factor of 1.84, and explains 80% of the variance in $N_{d0}$
amongst these cases.


### 7.2 Vertical profiles

Total particle number concentration data are block averaged in altitude bins of 500 m across all cloud free regions to capture
the vertical structure of the environment as sampled in situ by the Falcon. Figure 15 (panels b and c) shows the profile of
total particle number concentration (>3 nm, $N_3$), and $N_{100}$, as well as their ratio (panel d) and the non-volatile fraction, NVF
(>10 nm, panel e).

$N_3$ generally decreased or remained constant through the lowest 2 km and then increased above 3 km. This upward trend is
most apparent for the three cases conducted in 2022 from Bermuda (Cases 4-6) where a marked transition to higher
concentrations was observed and was also observed to occur for larger particles (>100 nm). In Cases 4 and 6, increasing $N_3$
with altitude above 3 km was correlated with enhancements in CO and $CH_4$ (Figure 13) and minor increases in particle
organic mass (Figure 14c), suggesting continental origin. While the beginnings of the uptick in particle concentrations were
consistent for Case 5, correlations were not discernible because the vertical extent of the sampling was truncated in line with
the vertical extent of the cloud cluster. The close vertical alignment of the increase in $N_3$ and $N_{100}$ in the environment
(Figure 15) with the inflection in the profile of $N_d$ for Cases 4B and 6 (Figure 7c) is perhaps an additional indication of
secondary activation of entrained CCN. Case 6 offers a favorable dataset to evaluate entrained activation (e.g., in parcel
models and LES) because of the sharpness of the aerosol gradient and the abundance of larger ($N_{100}$) particles aloft. In
contrast, Cases 2 and 3 show markedly reduced $N_{100}$ above 3 km, with a major distinction between these two cases occurring
between 2-3 km as a result of the organic-rich pollution layer affecting Case 3.

Across all cases the fraction of larger particles decreases with altitude (Figure 15d) with the ratio $N_{100} / N_3$ varying from 22-
40% in the lowest altitude bin to less than 6% above 4 km (for the cases where data was available). NVF provides an
indicator for refractory cores enhanced in layers enriched in primary combustion particles as well as dust and sea spray,
while low values typically indicate an abundance of (nucleated) secondary aerosol species. At low altitudes, the marine
dominated Cases 1 and 2 had the lowest NVF (32% and 42%, respectively) and is supportive of a large number fraction of





marine particles originating from new particle formation and subsequent secondary aerosol growth, with the pollution-influenced Case 3 the highest (54%), and the Bermuda 2022 Cases 4-6 tightly clustered between 48% and 50%. Near 2 km,

the influence of the continental pollution is most significant in Case 3, as determined by the organic enrichment (Figure 14c), and is accompanied by higher NVF, while Case 2 relaxes to an unperturbed free tropospheric background with low aerosol mass, smaller particles and the lowest NVF. It is notable that in the purportedly dust-influenced layer at 1.5-2.5 km influencing Cases 4-6, there is a distinct difference in NVF between Case 4 (which mimics the enhancements of Case 3) and Cases 5 and 6 (which instead decrease) and may provide further insight into the optical signature leading to the smoke

classification of Case 4 (Figure 13a).

## 8 Discussion and Conclusions

This measurement report describes six case studies relating to airborne observations of aggregated regions of marine shallow cumulus and cumulus congestus. The observations incorporate a coordinated flight strategy centered on a target cloud cluster and involved a HU-25 Falcon that sampled the cloud and surrounding environment in situ, while a King Air made repeated

remote sensing passes above the scene, dropping dropsondes around the perimeter and near the center.

The large-scale meteorology was broadly consistent across cases, with a subtropical anticyclone located to the east resulting in northward advection of moist tropical airmasses and the PBL airmass origin (8 day) was the trade wind region of the central tropical Atlantic. As the low-level airmasses advected around the southwest quadrant of the subtropical high moving northward into the sub-tropical region, mid-tropospheric temperatures decreased, while SSTs remained close to tropical

levels, typical of the summertime western Atlantic. Three of the cases were conducted over the warm Gulf Stream waters near the United States east coast, while the other three occurred over relative uniform SST near Bermuda. The warm, moist advection and surface fluxes maintained the PBL thermodynamic properties close to tropical conditions, while the reduction in mid-tropospheric temperature may explain the destabilization of the profile, permitting nearby deeper convection, and the lack of a defined capping inversion for most of the cases.

Overall, there was not a strong association between localized increases in static stability and the frequency of cloud top observations at that altitude. This may indicate that the environmental moisture profile, wind shear, cloud horizontal scale, and perhaps microphysics all contribute to the vertical distribution of cloud heights in situations where a strong capping layer is absent. Also notable was a lack of a universal relationship amongst the cluster motion vector, the shear vector and the principal linear axis of organization and is perhaps indicative of a memory effect that sustains a favored orientation after

formation.

Multiple passes through the cloud clusters revealed considerable variability in the cloud system properties compared to an idealized adiabatic parcel and confirmed the ubiquity of environmental entrainment in affecting the bulk characteristics. Evidence of transient thermals was observed through the variability in the spatial distribution of vertical velocity, bulk water content and microphysical properties between sequential transects. Downdrafts were commonly observed, often near cloud





edges, and there were some, albeit limited, examples where gas tracer concentrations within the cloud could only be explained by incorporating environmental air from a higher altitude. In some cases, cyclical collapse of emergent convective turrets resulted in net downward transport of cloud water near cloud top. These cases offer a unique dataset to evaluate and improve convective parameterization of entrainment and detrainment for shallow cumulus.

The effect of entrainment on generating sub-adiabatic $q_L$ was less impactful on $N_d$ and was found to be generally consistent
across cases. This was not associated with a uniformly reduced droplet growth rate (as might be anticipated with homogeneous mixing), but rather a source of $N_d$ from activation of entrained air and manifested as a distinct mode of small droplets in the DSD occurring alongside a mode that grew close to the (monodisperse) adiabatic rate. The result was a broadening of the DSD with altitude, as quantified by the relative dispersion. Further analysis and parcel modeling are needed to fully explore this concept in detail, but an underpinning factor is that the timescale for entrained air (e.g., at the
eddy scale of thermals) to homogenize within a cloudy volume is slow compared to the time taken for the entrained air to be carried upwards to its particular LCL. Further evidence that pointed towards the activation of entrained air was found in two cases where $N_d$ transitioned to an increase with altitude above 3 km in concert with an increase in the number of CCN-active particles in the environment.

Precipitation below cloud base was spatially concentrated into narrow regions and in some cases absent or negligible. It was
found in at least one case that rain water formed near cloud top was expelled laterally by subsequent thermals, where it was anticipated that much of the rain water would ultimately evaporate into the dry surrounding environment, explaining the limited extent of precipitation at lower altitudes. The cases which exhibited the highest near surface precipitation were associated with markedly different frequency distributions of cloud top altitude, perhaps indicating a threshold above which the precipitation substantially influences the cloud dynamics. The remaining cases would tend to indicate that precipitation
is a consequence, and not a cause, of aggregation and deepening.

Across the cases, shared attributes may provide useful means to isolate specific mechanisms or modes for further investigation. These are summarized as follows:

- Cases 1 and 5 were the least vertically developed and both exhibited pronounced linear organization with either very isolated or no observed precipitation below cloud base. Cloud motion was slightly (8˚ and 15˚, respectively) to
the left of the axis of cloud organization with veering of the wind (i.e., clockwise rotation) with altitude such that the cloud top to base shear vector was approximately perpendicular to the cloud axis and remarkably similar in magnitude. Different features for Case 1 included steering winds being stronger by a factor of ~2, being located over the Gulf Stream (rather than near Bermuda), and the aerosol was indicative of background marine conditions, while Case 5 was influenced by Saharan dust. Case 5 incorporated observations of two systems: 5A which was
decaying and 5B which was active, with 5A more influenced by the Saharan dust.

- Cases 2 and 3 exhibited intermediate levels of vertical development with a clear distinction between the vertically developed cluster and the surrounding boundary layer cumulus and both showed evidence of clearings forming in the immediate surroundings (more notable for Case 2). Both cases were located over the Gulf Stream. Case 3 was

more influenced by continental pollution leading to higher $N_d$. Convective vertical velocity extrema were stronger in Case 3, perhaps indicating a liquid-phase microphysical invigoration mechanism, while precipitation was more influential for Case 2. A notable difference in the organization was in the orientation of the cloud axis which was parallel to the cluster motion in Case 2 and perpendicular in Case 3.

- Cases 4 and 6 were cumulus congestus cases and the most vertically developed. The vertical distribution of cloud cover was different from the other cases, with the emergence of detrained layers in the environment surrounding the
convection. These cases also captured a distinct lifecycle, with the cloud cluster apparently deepening too rapidly to sustain itself. This was captured in the measurements of Case 4A, which sampled the remnants of a previously deeper system and in Case 6, where the cloud system ceased to exist shortly after the sampling was completed. These cases offer unique datasets for detailed analysis of convective lifecycle of cloud clusters.

In conclusion, this measurement report documents a novel strategy of utilizing two coordinated aircraft to conduct in situ and
remote sensing observations of aggregated shallow cumulus and cumulus congestus during the NASA EVS-3 ACTIVATE summer field campaigns spanning 2020-2022. Six process study cases have been reported, providing the background and supporting information associated with the measurements to guide further analysis and modeling.

**Data Availability**

All datasets are publicly available and can be found at https://doi.org/10.5067/SUBORBITAL/ACTIVATE/DATA001
(NASA/LaRC/ASDC, 2021).

**Video Supplement**

Satellite animations showing each case study on a cloud-centric moving coordinate system can be found at https://doi.org/10.5067/ASDC/SUBORBITAL/ACTIVATE-Satellite_1 (NASA/LaRC/ASDC, 2021).

**Author Contribution**

EC, LDZ, MAS, TS, JWH, AS, RAF designed and implemented the flight strategy. All authors contributed to experimental data collection and EC, LDZ and MAS conducted data analysis. EC led the preparation of the manuscript with contributions from all authors.

**Competing Interests**

At least one of the authors is a member of the editorial board of Atmospheric Chemistry and Physics.



**Acknowledgements**

We acknowledge the contributions from the pilots and aircraft support personnel from the NASA Langley Research Services Directorate for the successful execution of ACTIVATE flights. This work was supported by ACTIVATE, a NASA Earth Venture Suborbital (EVS-3) investigation funded by NASA's Earth Science Division and managed through the Earth System Science Pathfinder Program Office. CV and SK acknowledge support from the German Research Foundation.

**Financial Support**

Armin Sorooshian was supported by NASA (grant no. 80NSSC19K0442). Christiane Voigt and Simon Kirschler were funded by funded by the German Research Foundation DFG by SPP 1294 HALO under contract VO 1504/7-1 and VO1504/9-1 and by TRR 301 – Project-ID 428312742.

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



Table 1: Cloud cluster sampling characteristics, lifecycle and motion for Cases 1-6.

| | | | **1** | **2** | **3** | **4** | | **5** | | **6** |
|---|---|---|---|---|---|---|---|---|---|---|
| | | | | | | (A) | (B) | (A) | (B) | |
| | | | 9/29/2020 | 6/2/2021 | 6/7/2021 | 6/10/2022 | | 6/11/2022 | | 6/14/2022 |
| Module Midpoint | Time | (UTC) | 15:54 | 19:00 | 19:07 | 19:13 | 20:11 | 18:32 | 19:49 | 15:05 |
| | Latitude | (°) | 37.80 | 34.41 | 36.31 | 31.71 | 31.17 | 30.38 | 31.09 | 30.45 |
| | Longitude | | -70.11 | -74.83 | -73.60 | -65.56 | -65.96 | -65.54 | -64.40 | -64.31 |
| Duration | King Air | (hr) | 1.2 | 1.4 | 1.6 | 3.3 | - | 0.8 | 1.5 | 2.5 |
| | Falcon | | 0.8 | 0.9 | 1.4 | 1.7 | 0.8 | 0.5 | 1.5 | 2.2 |
| Fitted cloud drift velocity | U | $ms^{-1}$ | 0.96 | -0.78 | 3.12 | 5.29 | 5.40 | 0.74 | 1.15 | 3.98 |
| | V | | 7.40 | 7.61 | 4.32 | 3.29 | 2.28 | 3.76 | 2.88 | 3.28 |
| | dU/dt | $ms^{-1}\ hr^{-1}$ | 0.22 | 0.52 | 0.18 | -0.16 | -0.58 | 0.51 | 0.30 | 1.10 |
| | dV/dt | | 0.19 | 0.50 | 0.64 | -0.16 | -0.49 | -0.45 | -0.56 | 0.34 |
| Lifecycle | Start | (UTC) | 12:40 | 13:40 | 14:00 | 16:20 | 18:00 | 17:40 | 17:40 | 12:20 |
| | End | | 18:00 | 21:20 | 20:40 | 21:00 | 21:00 | 19:00 | 21:00 | 16:20 |
| | Lifetime | (hr) | 5.3 | 7.7 | 6.7 | 4.7 | 3 | 1.3 | 3.3 | 4 |




**Table 2: Cloud and environmental properties of Cases 1-6.**

| | | | 1 | 2 | 3 | 4 A | 4 B | 5 A | 5 B | 6 | Source[a] | Notes |
|---|---|---|---|---|---|---|---|---|---|---|---|---|
| Cloud Fraction | | - | 0.47 | 0.38 | 0.23 | 0.42 | | 0.095 | 0.26 | 0.33 | K | HSRL cloud top detection |
| Mixed layer height | | m | 290 | 490 | 370 | 100 | | 290 | 220 | 350 | D | First max in RH (see text) |
| Lowest cloud base | | m | 285 | 530 | 413 | 1100 | 325 | 340 | 225 | 440 | F | Falcon cloud detection and camera |
| Height of max. cloud top frequency | | m | 510 | 790 | 650 | 2860 | - | 460 | 390 | 2520 | K | HSRL cloud top detection |
| Sea Level Pressure | | hPa | 1013.7 | 1019.8 | 1019.9 | 1016.9 | | 1019.4 | 1019.7 | 1019.7 | D | |
| Mixed Layer | $\theta$ | K | 298.3 ±0.3 | 296.5 ±0.15 | 297.7 ±0.23 | 296.3 ±0.18 | | 297.3 ±0.12 | 296.8 ±0.16 | 297.6 ±0.09 | D | |
| | $q_v$ | g kg$^{-1}$ | 17.7 ±0.40 | 14.2 ±0.42 | 16.3 ±0.35 | 17.4 ±0.15 | | 16.9 ±0.34 | 17.1 ±0.23 | 16.3 ±0.23 | D | |
| | U | m s$^{-1}$ | 0.4 ±0.6 | -2.6 ±0.7 | 3.7 ±0.7 | 2.5 ±0.6 | | 0.6 ±0.5 | 0.8 ±0.6 | 2.5 ±0.6 | D | |
| | V | m s$^{-1}$ | 8.1 ±1.4 | 7.8 ±0.9 | 3.9 ±1.0 | 4.8 ±0.8 | | 3.9 ±0.5 | 3.9 ±0.5 | 2.5 ±0.9 | D | |
| | $\sigma_w$ | m s$^{-1}$ | 0.5 | 0.7 | 0.6 | 0.3 | 1.0 | 0.5 | 0.5 | 0.6 | F | Turbulent winds |
| LWP | | g m$^{-2}$ | 936 | 1480 | 2680 | 629 | 2340 | 1410 | 2650 | 3360 | F | Integrated mean water content |
| PW | | kg m$^{-2}$ | 45.3 | 37.7 | 35.4 | 47.4 | | 45.7 | 48.3 | 41.8 | D | Eq 1 applied to mean dropsonde $q_v$ |
| $N_{d,0}$ | | mg$^{-1}$ | 144 | 175 | 508 | 315 | 301 | 212 | 302 | 170 | F | adiabatic cloudbase estimate (see text) |
| CCN 0.37% | | mg$^{-1}$ | - | 197 | 437 | 513 | 481 | 256 | 380 | 193 | F | |
| $N_1$ | | mg$^{-1}$ | 179 | 197 | 908 | 412 | 417 | 182 | 258 | 159 | F | Fitted lognormal distribution parameters |
| $D_1$ | | nm | 43 | 47 | 64 | 55 | 62 | 56 | 64 | 51 | F | |
| $\sigma_1$ | | - | 0.17 | 0.18 | 0.24 | 0.19 | 0.19 | 0.19 | 0.16 | 0.18 | F | |
| $N_2$ | | mg$^{-1}$ | 86 | 149 | 106 | 187 | 160 | 123 | 135 | 100 | F | |
| $D_2$ | | nm | 198 | 186 | 231 | 180 | 189 | 194 | 190 | 179 | F | |
| $\sigma_2$ | | - | 0.16 | 0.16 | 0.11 | 0.15 | 0.13 | 0.13 | 0.09 | 0.14 | F | |
| $D_{act}$ | | nm | 52 | 74 | 69 | 68 | 74 | 57 | 56 | 54 | F | |
| AOD | | - | 0.093 | 0.064 | 0.096 | 0.16 | | 0.14 | 0.1 | 0.1 | K | HSRL extinction |
| w [max/min] | | m s$^{-1}$ | -5.22, 5.29 | -2.88, 4.35 | -5.70, 10.9 | -1.24, 1.01 | -3.29, 6.83 | -3.53, 2.08 | -3.88, 5.79 | -5.53, 8.39 | F | |
| CAPE | pseudo adiabat | J kg$^{-1}$ | 960 | 228 | 612 | 471 | | 757 | 703 | 650 | D | using mixed layer properties above |
| | reversible | | 217 | - | 33 | 40 | | 66 | 51 | 31 | D | |
| CIN | | J kg$^{-1}$ | 3 | 3 | 5 | 3 | | 3 | 2 | 5 | D | |



| | | | | | | | | | | |
|---|---|---|---|---|---|---|---|---|---|---|
| LCL | m | 426 | 639 | 501 | 222 | | 410 | 342 | 546 | D | |
| SST | K | 301.1 | 299.4 | 300.9 | 297.2 | 297.7 | 298.9 | 298.3 | 300.0 | F | KT-15 |
| SHF | W m$^{-2}$ | 4 | -1 | 2 | 3 | - | -7 | -1 | -1 | F | eddy covariance at minimum altitude |
| LHF | W m$^{-2}$ | 159 | 122 | 38 | 24 | - | 40 | 18 | 72 | F | eddy covariance at minimum altitude |
| CO | ppb$_v$ | 86 ± 4 | 83 ± 3 | 90 ± 7 | 85 ± 3 | 86 ± 3 | 80 ± 3 | 81 ± 5 | 72 ± 3 | F | module average concentration |
| CO2 | ppm$_v$ | 409 ± 0.3 | 419 ± 0.2 | 419 ± 0.7 | 421 ± 0.2 | 420 ± 0.3 | 420 ± 0.2 | 420 ± 0.2 | 420 ± 0.2 | F | |
| CH4 | ppb$_v$ | 1914 ± 4 | 1908 ± 1 | 1925 ± 6 | 1922 ± 4 | 1925 ± 8 | 1916 ± 2 | 1922 ± 12 | 1918 ± 4 | F | |
| O3 | ppb$_v$ | 28 ± 7 | 28 ± 11 | 34 ± 17 | 24 ± 6 | 31 ± 14 | 17 ± 4 | 19 ± 4 | 23 ± 12 | F | |
| Max Rain Intensity | mm hr$^{-1}$ | - | 1.55 | 0.97 | 2.58 | 4.29 | - | 3.87 | 4.41 | F | |
| Mean Rain Intensity | mm hr$^{-1}$ | - | 0.47 | 0.35 | 0.97 | 1.41 | - | 1.11 | 1.53 | F | |
| Rain coverage | km | - | 7.1 | 0.7 | 17.1 | 9.2 | - | 0.6 | 10.4 | F | |
| Fractional rain coverage | - | - | 0.24 | 0.02 | 0.47 | 0.36 | - | 0.02 | 0.17 | F | |
| Cluster Rain Rate | mm hr$^{-1}$ | - | 0.11 | 0.007 | 0.45 | 0.52 | - | 0.02 | 0.26 | F | |

a Source:K = King Air observations, D = Dropsonde, F = Falcon observations




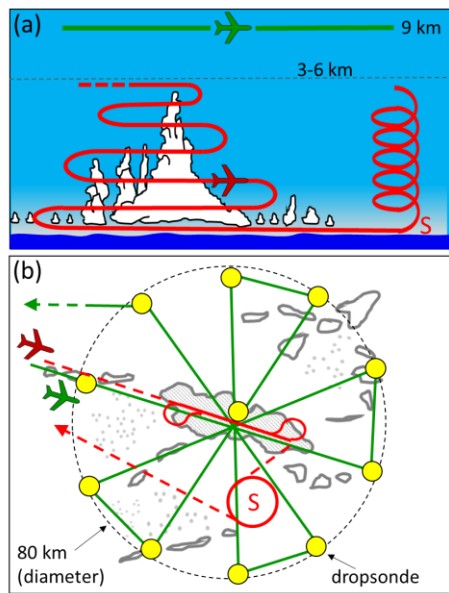

Figure 1: Schematic of the nominal two aircraft flight strategy illustrating the position of the King Air (green) and the Falcon (red): (a) Cross section view of the altitude profile flown through the target cloud with adjacent clear spiral (S), and (b) plan
view showing the "wheel and spoke" pattern.





Figure 2: Large-scale meteorological environment. (a) 8-day back trajectory (solid) and 1-day forward trajectory (dashed) using MERRA-2 800-950 hPa layer averaged winds (see text) for Cases 1-6. (b) Example SST shown for Case 6 (2022-06-14) with all case locations shown for reference. The region in (b) corresponds to the grey shaded region of (a). (c) Case 6: sea-level pressure (contours), 850 hPa wind vectors, and 700 hPa temperature (colored). The location of the axis of the stationary front (see text) is shown as the thick red dash. (d) Case 6: 500 hPa geopotential height (contours at 6 dm intervals, thick contour designates the 588 dm level), 1000-500 hPa thickness (dashed contours at 6 dm intervals, black contour designates the 564 dm thickness line, red (blue) contours represent regions of higher (lower) thickness), PW (colored), and MF (vectors).





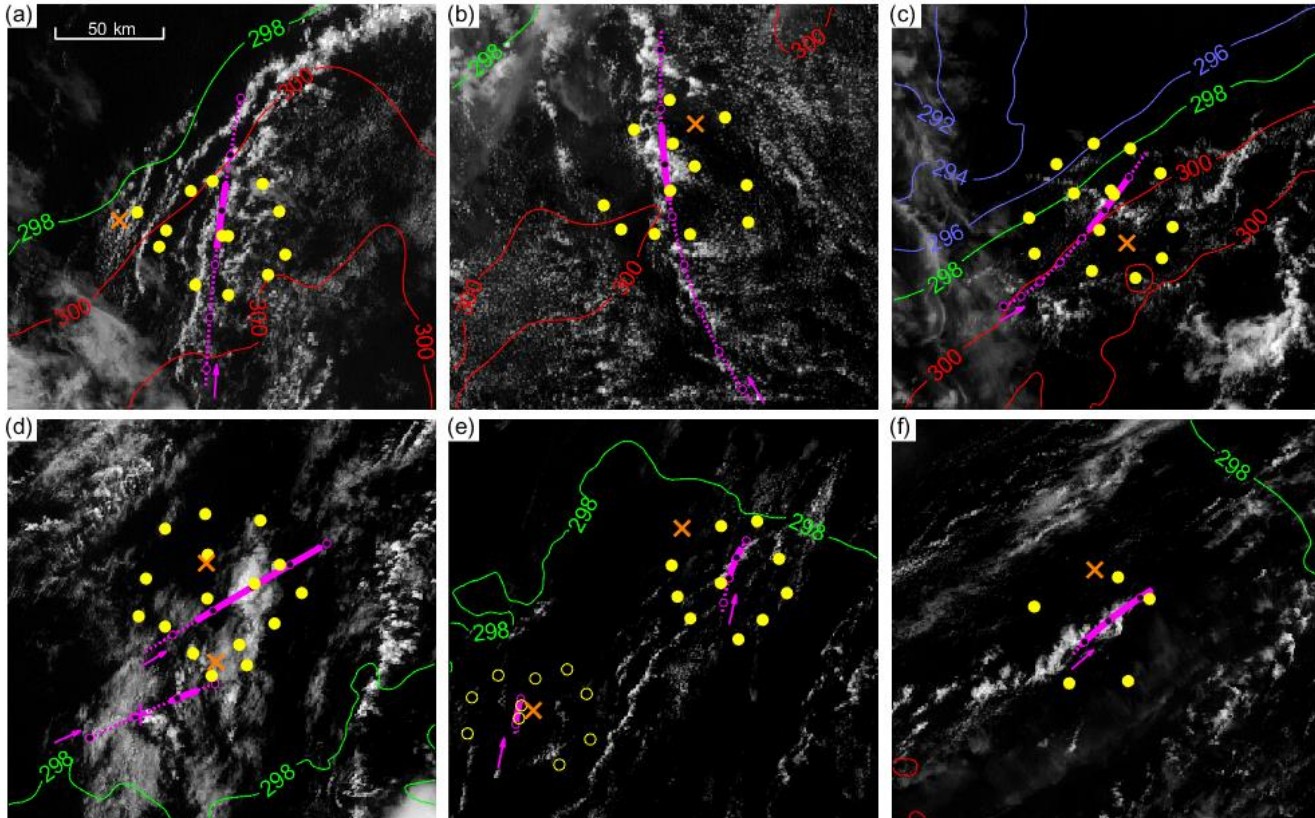

Figure 3: GOES-East ABI visible (0.6 µm) satellite imagery at the midpoint time of each process study: (a)-(f) Case 1-6,
respectively, with a scale bar shown in panel (a). Overlaid in each panel: SST (K, contours), dropsonde locations (yellow
dots), Falcon clear spiral location (orange cross), cloud cluster track (magenta). The cloud cluster track (see text) is
separated into the timeframe of aircraft sampling (solid) and the remaining time window of the cluster lifetime (dotted).
Hourly increments are shown as circles and an adjacent arrow shows the direction of travel. (d) and (e) are images from the
midpoint of Case 4A and 5B, respectively. No dropsondes were associated with Case 4B, but its cloud track is shown to the
southwest of Case 4A; the location of the tracked Case 4B cluster at the time of the panel (d) image is marked with a plus
symbol (+). Dropsondes associated with Case 5A are shown as open circles in panel (e) and the tracked cluster had already
dissipated.





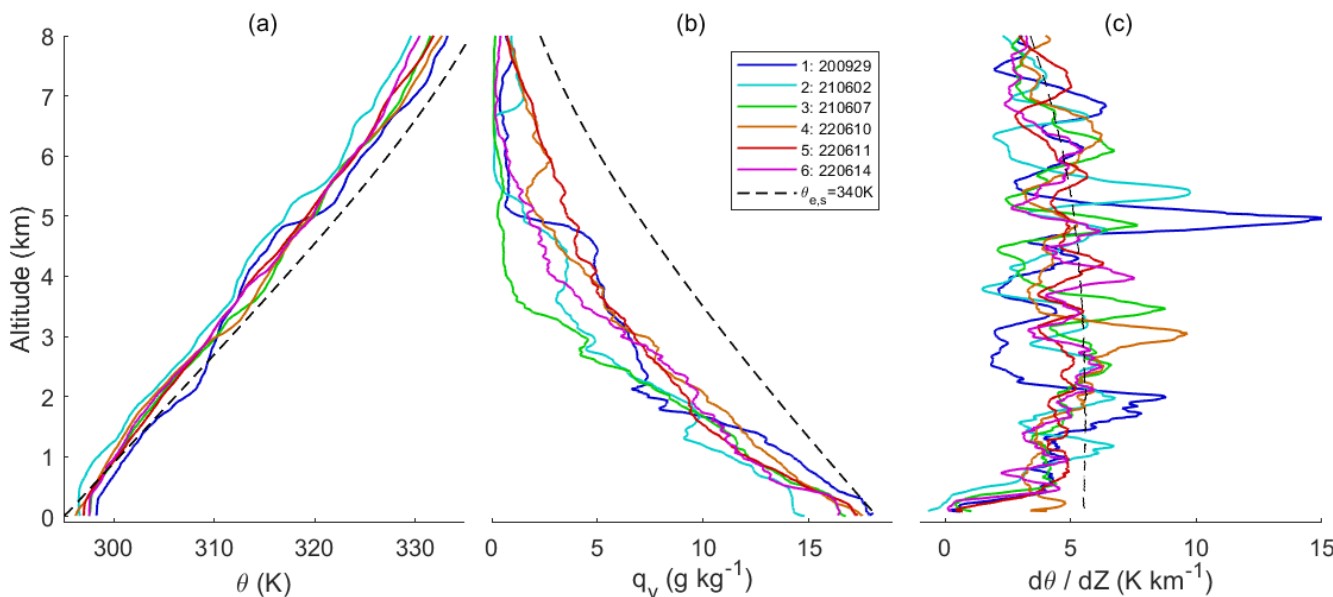

Figure 4: Case-mean dropsonde profiles of (a) potential temperature, (b) water vapor mixing ratio, and (c) static stability (defined as the vertical gradient of potential temperature and smoothed over a 100 m window). The profile of a reference wet adiabat corresponding to a saturation equivalent potential temperature ($\theta_{e,s}$) at 340 K is also shown.



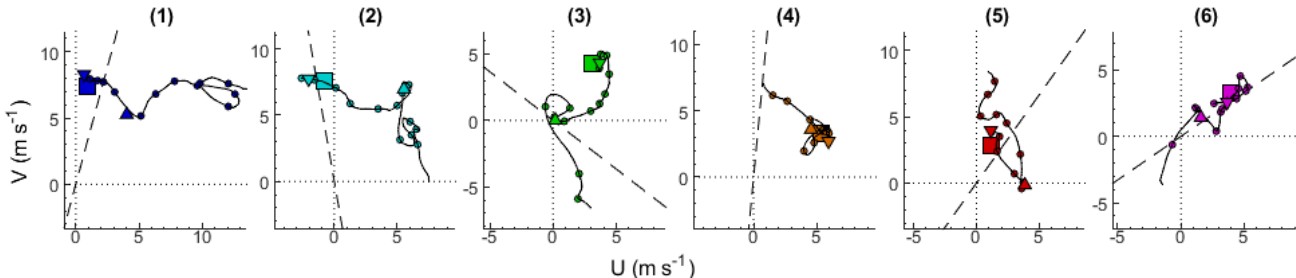

Figure 5: Case-mean dropsonde wind profiles displayed as hodographs for Cases 1-6. The wind profile is smoothed with a 25 hPa running mean and markers (o) indicate increments of 50 hPa. Also included are winds at cloud base (down triangle) and cloud top (up triangle, estimated based on the top of the Falcon module), the mean cluster motion from satellite (square), and the principal axis of cloud organization assessed from satellite (dashed line). In each panel, the scale is preserved with

1240 the position of the origin shifted to center the data. Case 5A dropsondes are omitted from (5) for readability but indicate similar structure to Case 5B (shown).





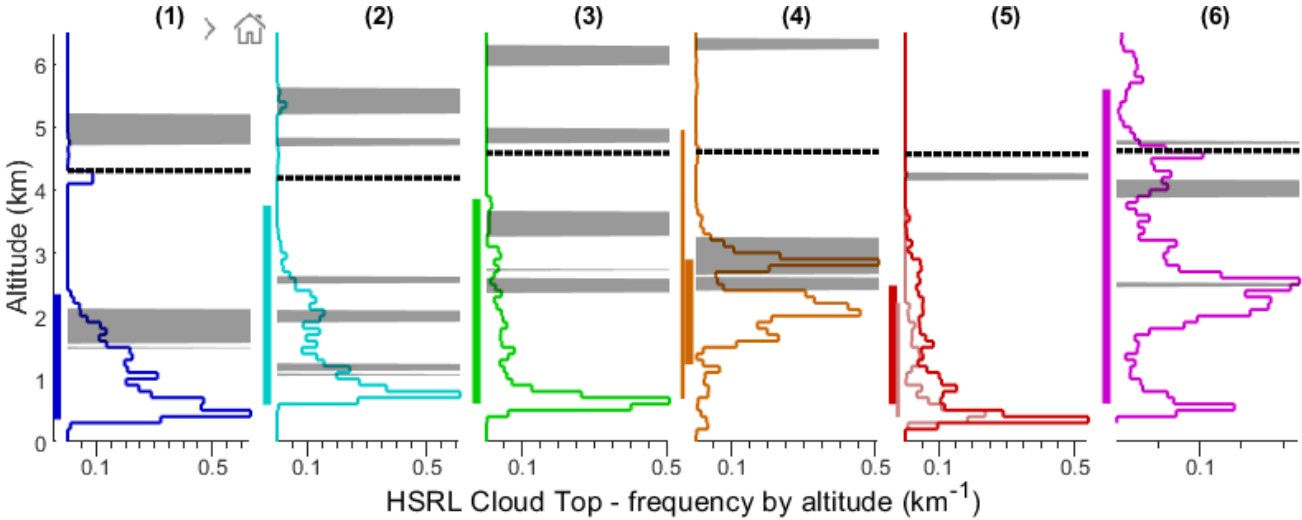

Figure 6: Frequency distributions of HSRL detected cloud tops for Cases 1-6. Frequency data are normalized by the total available records within the module, such that an integration of the distribution results in the HSRL cloud fraction (Table 2). Also shown are regions where dθ/dz exceeds 6 K km$^{-1}$ as an indicator of stable layers (grey shading), and the 0˚C level (dashed). The vertical bar adjacent to each case indicates the span of altitudes sampled in situ by the Falcon, and in panel (4) and (5), the altitude span for the secondary Falcon modules (thin bar, Cases 4B and 5A, respectively).





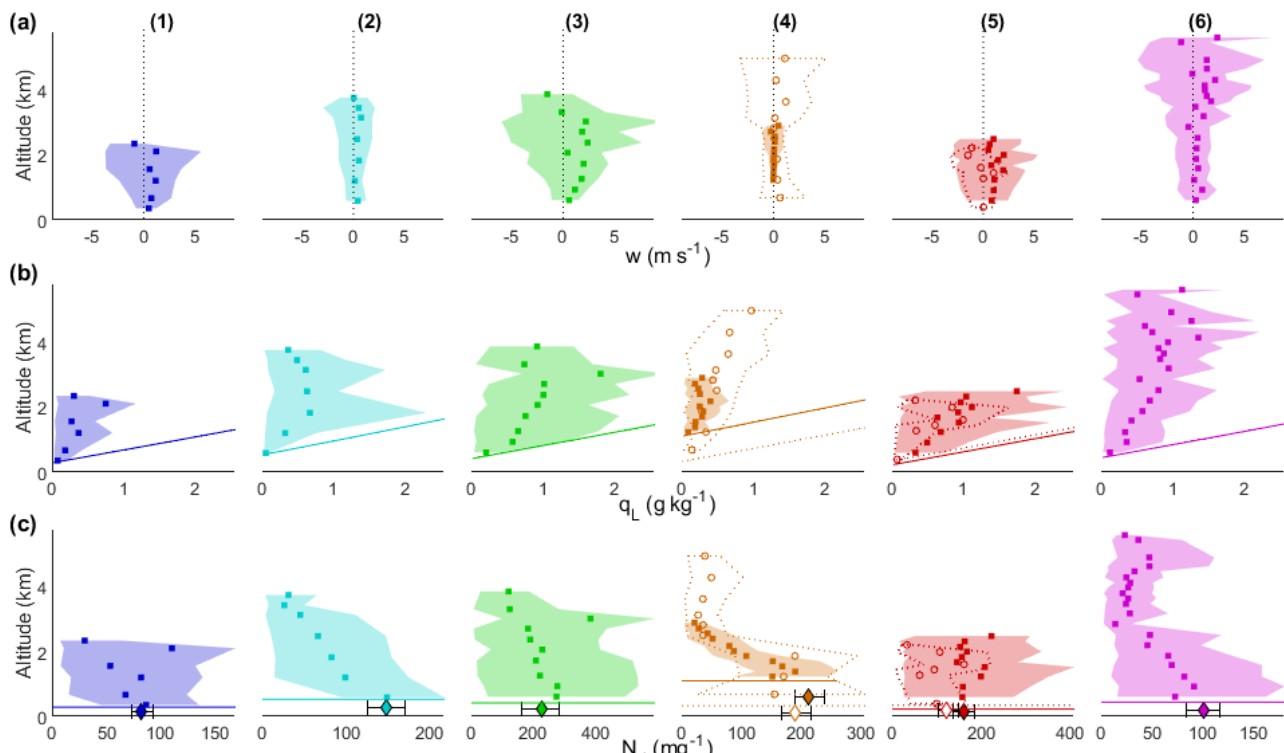

Figure 7: Statistics of in-cloud properties, measured in situ for Cases 1-6: (a) vertical velocity, w, (b) liquid water mixing ratio, $q_L$, (c) drop number concentration, $N_d$. The shaded region corresponds to the 10-90% range and dots are the transect mean values where data are filtered for cloud using $q_L > 0.02$ g kg$^{-1}$. In the case of (a), the transect means are calculated as a $q_L$-weighted mean, referred to as $w_L$ in the text. Also shown in (b) is an adiabatic parcel initialized from saturation at the lowest cloud base, shown as a horizontal line in (c). Also shown in (c) is a reference aerosol concentration based on the particle number concentration, exceeding 100 nm diameter, measured during sub-cloud sampling (marker: mean, bar: 10-90% range). In (4) and (5), data for the secondary modules are included (Case 4B and 5A, respectively) with dashed lines and open markers.



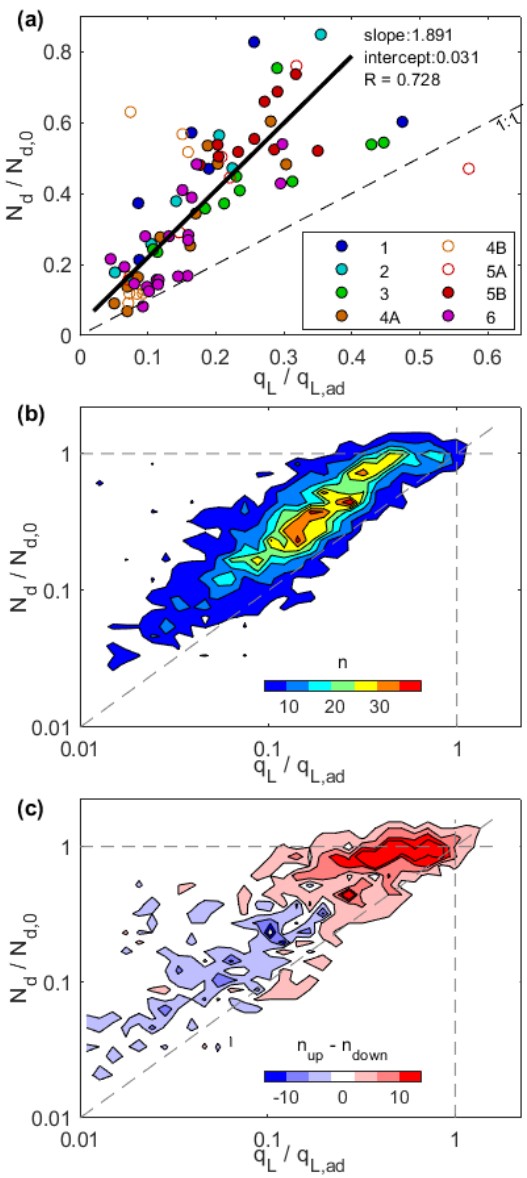

Figure 8: Variation of $N_d$ and $q_L$ with respect to a reference parcel with properties $N_{d,0}$ and $q_{L,ad}$. (a) Transect in-cloud mean for Cases 1-6 (open circles for secondary modules). A linear model is fit to the data using total least squares with equal uncertainty in $q_L$ and $N_d$. (b) Joint frequency distribution (counts, n) of all cloudy data for all cases with logarithmically spaced bins. (c) Same as (b) but showing the difference in the joint frequency distribution between updrafts and downdrafts.





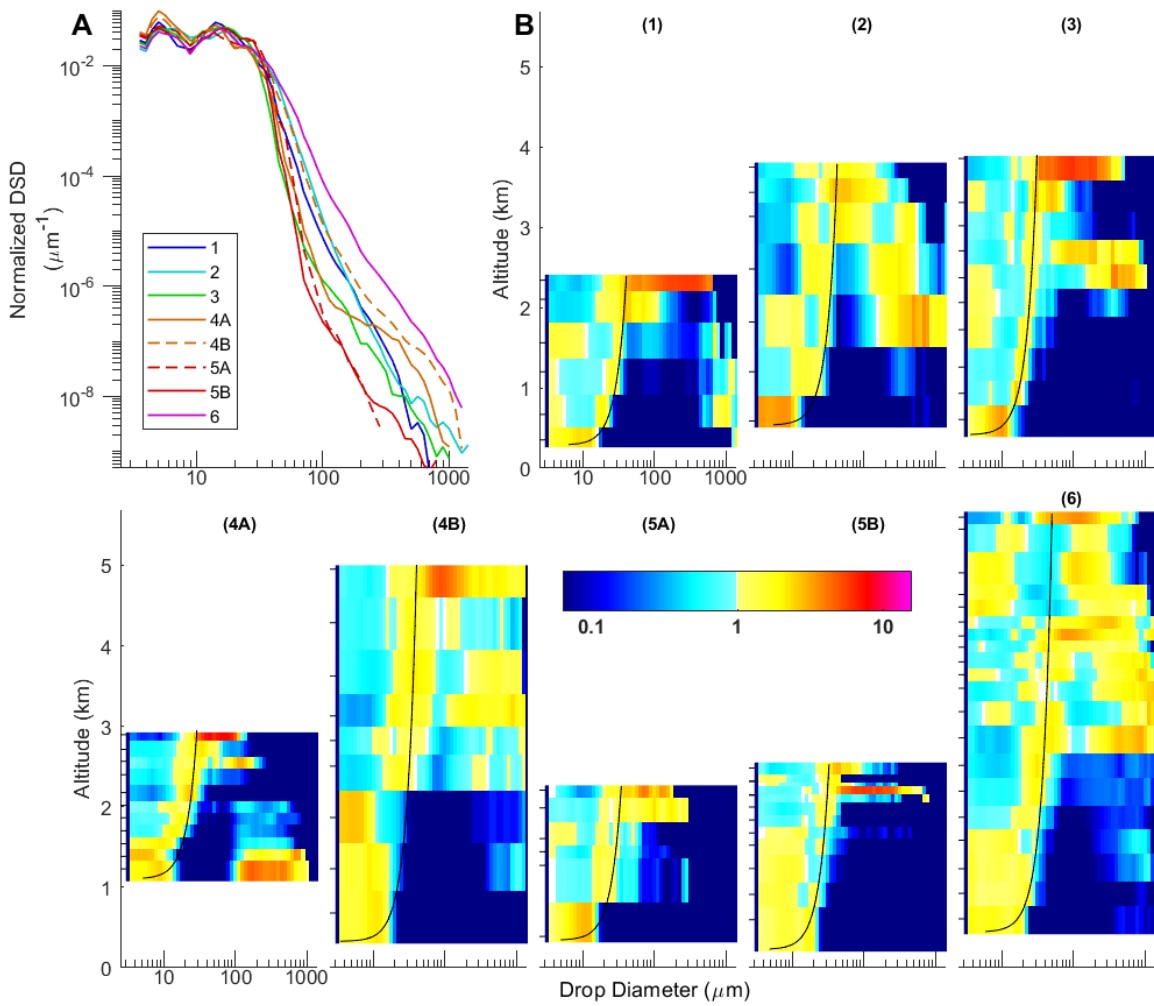


Figure 9: Vertical profile of the anomaly normalized drop size distributions (DSD). (A) Reference normalized DSD calculated from the mean across all levels sampled. (B) For Cases 1-6: At each level, the transect mean in-cloud DSD is normalized to unit integral, and then the anomaly is calculated as a ratio to the reference DSD. Warm (cold) colors therefore represent higher (lower) than average contribution to the DSD shape and use of the anomaly permits comparison across the size spectrum, despite the large change in concentration. Also shown for each case is the monodisperse drop diameter of an adiabatic parcel (black line) initialized at the lowest cloud base with a drop number concentration, $N_0$ (see text).



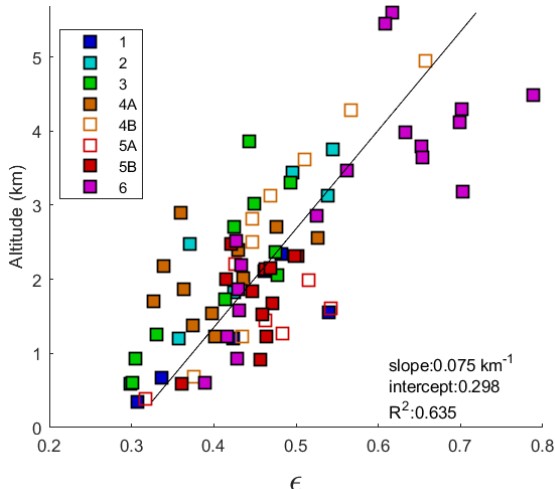

Figure 10: Profiles of DSD relative dispersion, $\epsilon$, for Cases 1-6. Values of $\epsilon$ are calculated locally at the measurement

interval (1s, ~100 m) and then averaged (weighted by $N_d$) across a transect.





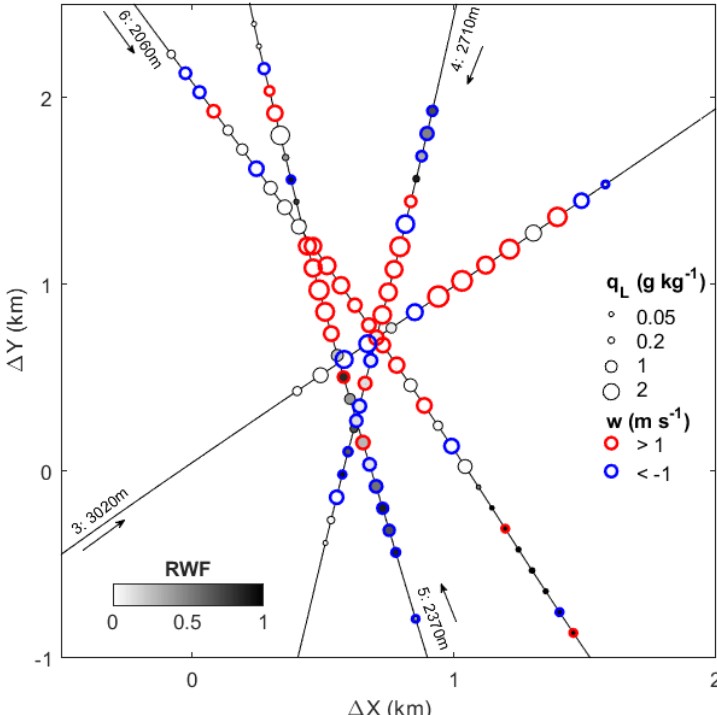

Figure 11: Four sequential Falcon in situ transects (2-3 km altitude) through cloud during Case 3. The aircraft position has been projected onto a cloud-centric coordinate where X (Y) represents the displacement along (perpendicular to) the cluster motion. The size of the markers corresponds to the total $q_L$ with shading indicating the rainwater fraction (RWF) defined as the fraction of $q_L$ resulting from drop diameters exceeding 100 µm. Red (blue) rings indicate updrafts (downdrafts).



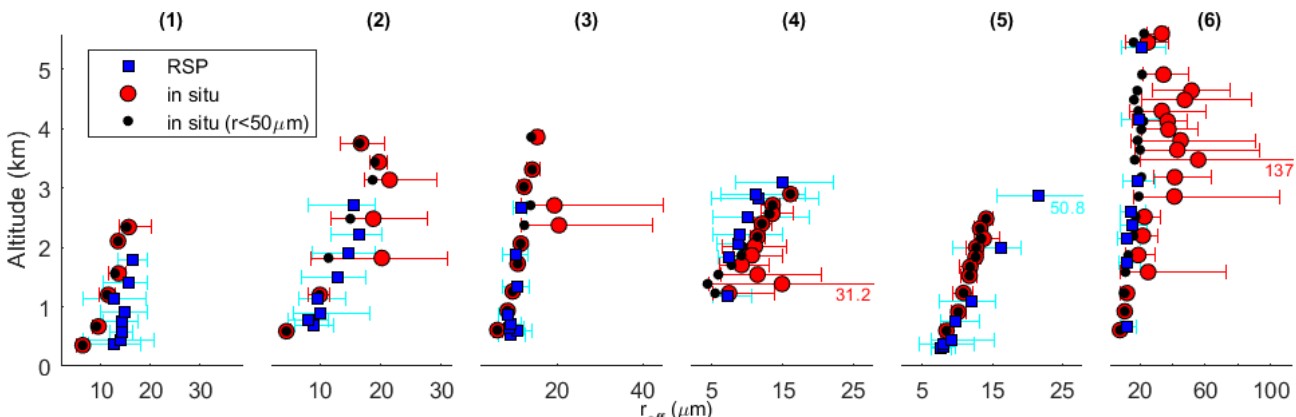

Figure 12: Comparison of the vertical profile of effective radius, $r_{eff}$, for Cases 1-6 between Falcon in situ cloud transects and
composite profiles from RSP. The composite profiles were derived by grouping RSP retrievals into eight equal frequency
bins of cloud top height as determined by HSRL, with the mean altitude of each bin displayed. A comparison is made with in
situ derived $r_{eff}$ omitting contributions from rain water (r < 50 µm). For both datasets, the mean (markers) and 10-90% range
(bars) are shown.





Figure 13: Trace gas vertical profiles separated into a far-field component measured during the Falcon clear spiral (black), a near-field component measured outside but in the vicinity of the cloud cluster at cloudy altitudes (green), an in-cloud component (blue), and a sub-cloud component (cyan). Data are shown for Cases 1-6 for (a) CO, (b) $CO_2$, (c) $CH_4$, and (d) $O_3$. Note the changes in concentration scales.









Figure 14: Vertical profiles of aerosol and cloud composition. (a) aerosol type frequency as derived by HSRL, (b) aerosol extinction components and comparison with HSRL, (c) aerosol sub-micrometer mass measured by the AMS, and (d) ionic composition of cloud water (note that no samples were captured during Case 5A and repeated samples at the same level are marked).




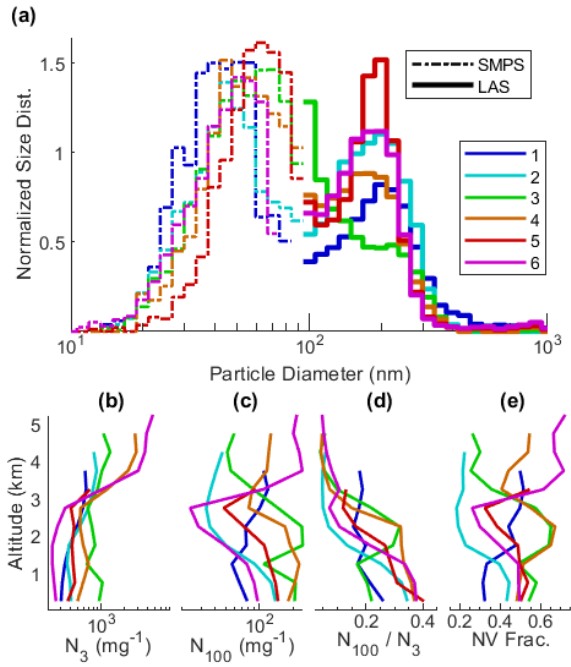

Figure 15: Aerosol microphysical properties for each case. (a) Normalized particle size distributions in the sub-cloud region, profiles of (b) total particle number concentration (>3 nm), (c) total particle number concentration (>100 nm), (d) the number fraction >100 nm, and (e) non-volatile number fraction (>10 nm).