# Peer review of "Measurement Report: Cloud and environmental properties associated with aggregated shallow marine cumulus and cumulus congestus"

_EGUsphere, 2024_

## Referee Comment (RC1)

**Review of "Measurement Report: Cloud and environmental properties associated with aggregated shallow marine cumulus and cumulus congestus" by Ewan Crosbie et al.**

This measurement report details six marine shallow cumulus cases sampled during the NASA ACTIVATE field campaign during summer months over the western Atlantic Ocean in a comprehensive, inclusive, and scientifically intriguing way. Besides the summary of the aircraft sampling approach and individual measurements, a number of interesting observations regarding shallow convective cumuli in this region are highlighted: 1) there is a lack of correlation between static stability and cloud-top frequency; 2) cluster motion vector does not seem to have an apparent relationship with the shear vector nor the axis of organization; 3) the impact of entrainment on the subadiabaticity of $q_L$ is stronger than that of $N_d$; 4) cloud DSD broadens with altitude. The report covers a wide range of scientific topics that these measurements can shed light on, including thermodynamical, dynamical, and microphysical features of shallow cumulus. Variations as well as shared attributes across the cases are discussed.

The manuscript is really well written. Individual measurements are clearly described and discussed in scientific contexts. This report should be of great interest to others in the community and serves as foundations for future topic-oriented studies making use of these comprehensively sampled cumulus cases. I think this manuscript is worthy of speedy publication to raise community awareness of the potential in these measurements.

I only have the following minor comments:

- Mesoscale organization, self-aggregation of convective features and aerosol-cloud interactions are the main themes in the introduction, but I did not see much discussion along these lines in the results. Could be worthwhile adding a few statements touching base on these topics.

- Line 135, "Methods" and Data?

- Line 143, could you comment on the representativeness of these cases on the general features of shallow Cu in this region?

- Cloud DSD (lines 197-205), could you add at what Hz is individual measurement taken and the temporal resolution after the two DSDs from FCDP and 2DS are stitched?

- Lines 216-230, similar to the above comment, I think the readers would appreciate the information on the temporal resolution of these measurements (and after stitching).

- Line 295-299, For this cloud/feature tracking method, is there any assumption involved (e.g., decorrelation timescale, threshold for the cross-correlation), in order words, how did you make sure it's still the same cloud/feature that you are tracking (the ~8hr cumulus lifetime in case 2 seems a little suspicious to me in this regard).

- Line 324-325, This is an interesting observation, any speculation on why? Stability? Shear?

- Line 335-336, "near-parallel linear cloud features (cloud streets)" and "near-perpendicular shear vector", are these two features physically connected or just by coincident, I think the reader will be curious.

- Line 351, reference for this derivation of mixed-layer height?

- Line 363, reference for using 6 K km$^{-1}$ to indicate stable layers?

- Figure 6, wouldn't it be nicer to be able to see LCL (or MLH) directly from the graph? (also check panel (1) for misprinting).

- Line 499-503, not sure if I follow this argument, how does spatial scale come in to play here, may need to clarify.

- Discussion and Conclusions, you already touched on this by providing a bullet list of interests for future investigations, but I think it's worth emphasizing/mentioning the strengths of this dataset (i.e., rich in detail and high spatiotemporal resolution that will shift the focus away from statistical description of features in an ensemble sense) in the context of stimulating future topic-oriented studies.

---

## Author Comment (AC1)

Response to reviewers

We thank the two reviewers for their helpful comments. We have provided our responses to comments below in blue.

Reviewer 1:

**Review of "Measurement Report: Cloud and environmental properties associated with aggregated shallow marine cumulus and cumulus congestus" by Ewan Crosbie et al.**

This measurement report details six marine shallow cumulus cases sampled during the NASA ACTIVATE field campaign during summer months over the western Atlantic Ocean in a comprehensive, inclusive, and scientifically intriguing way. Besides the summary of the aircraft sampling approach and individual measurements, a number of interesting observations regarding shallow convective cumuli in this region are highlighted: 1) there is a lack of correlation between static stability and cloud-top frequency; 2) cluster motion vector does not seem to have an apparent relationship with the shear vector nor the axis of organization; 3) the impact of entrainment on the subadiabaticity of qL is stronger than that of Nd; 4) cloud DSD broadens with altitude. The report covers a wide range of scientific topics that these measurements can shed light on, including thermodynamical, dynamical, and microphysical features of shallow cumulus. Variations as well as shared attributes across the cases are discussed.
The manuscript is really well written. Individual measurements are clearly described and discussed in scientific contexts. This report should be of great interest to others in the community and serves as foundations for future topic-oriented studies making use of these comprehensively sampled cumulus cases. I think this manuscript is worthy of speedy publication to raise community awareness of the potential in these measurements.

I only have the following minor comments:

Mesoscale organization, self-aggregation of convective features and aerosol-cloud interactions are the main themes in the introduction, but I did not see much discussion along these lines in the results. Could be worthwhile adding a few statements touching base on these topics.

Line 135, "Methods" and Data?

Revised, thank you.

Line 143, could you comment on the representativeness of these cases on the general features of shallow Cu in this region?

The collection of features, organization patterns, scales and general cloud structure captured across the six process study cases is representative of a subset of shallow Cu environments across this region in the summertime.  Quantifying the prevalence of modes that resemble those described here has not yet been done, notwithstanding our anecdotal understanding built from trying to predict their whereabouts during the field campaign.  It is challenging to succinctly revise the text to communicate this, but we have added the following to L145:

"…cumulus congestus, which was a regularly occurring cloud pattern observed across the region."

Cloud DSD (lines 197-205), could you add at what Hz is individual measurement taken and the temporal resolution after the two DSDs from FCDP and 2DS are stitched?

Both are acquired at 1 Hz and stitching is done at 1 Hz. Analysis performed on the size distributions usually involved integration/averaging over a cloud transect, but the merge was created at the timescale of acquisition. The sampling interval has been added to the text.

Lines 216-230, similar to the above comment, I think the readers would appreciate the information on the temporal resolution of these measurements (and after stitching).

The sample interval was already provided for the LAS and SMPS along with a reference (L218-219). This is the time resolution that the individual instrument datasets are archived. In this study, these data are reported as averages over several scans. This is intentionally left vague because each case has a different amount of time allocated to sampling the regions where averaging of the data occurs. The result is materially unaffected by the order of operations (e.g., stitch then average or average then stitch), so we feel the LAS/SMPS description is good as is.

The following is added to L224-225:
"The CPCs, CCN counter, and nephelometers all provided data at 1 Hz."

"at 30s intervals" is added to L225 in relation to the AMS.

Line 295-299, For this cloud/feature tracking method, is there any assumption involved (e.g., decorrelation timescale, threshold for the cross-correlation), in order words, how did you make sure it's still the same cloud/feature that you are tracking (the ~8hr cumulus lifetime in case 2 seems a little suspicious to me in this regard).

Hopefully the reviewer viewed the video supplements that were archived and linked to the paper in anticipation of this question. These are centered on the track so it should be obvious to the viewer whether a continuous region/cluster is being followed.

As a reminder, we are tracking a region of convective aggregation and not the attributes of a single thermal or cumulus cycle. That happens on a much shorter time scale and was a key focus of our analysis described in Section 5.2. Thermal cycles were constantly emerging and dissipating within the same complex. We cannot comment further on the root of the reviewer's suspicions about the cluster lifetime or attempt to assuage them.

We should also clarify that several different "human observer" approaches were taken in conjunction with the algorithmic approach and we are confident that it was satisfactory for this purpose. There was a scene correlation coefficient threshold of 0.2 that was imposed in helping to judge the end points of the track (i.e., when a coherent object could no longer be determined). Only sequential images were analyzed this way mainly because advection and cloud evolution were occurring simultaneously.
However, this threshold was also dependent on the scale of the selected region and the dominant cloud length scales within it, so it does not have broader significance. There were other factors such as the overrunning mid-and high cloud layers mentioned in the text that also complicated its use. Since our intention was not to develop a generalizable tracking algorithm, no further discussion of these aspects is included to avoid distraction from the main goals.

Line 324-325, This is an interesting observation, any speculation on why? Stability? Shear?

We agree with the reviewer that this was a notable finding but do not wish to speculate further at this time. This aspect is still an area of investigation and hopefully this paper can encourage interested parties to engage in modeling studies and potentially steer future targeted observational strategies.

Line 335-336, "near-parallel linear cloud features (cloud streets)" and "near-perpendicular shear vector", are these two features physically connected or just by coincident, I think the reader will be curious.

Our expectation had been to find shear vectors aligned with cloud streets, i.e., with minimal directional change. We found it notable that the two cases that exhibited the most linear organization shared this unexpected shear characteristic. Drawing particular attention to it was suggestive that it may not be coincidental. Hopefully readers are curious (see above response).

Line 351, reference for this derivation of mixed-layer height?

No reference is needed. The method we used was explained here; it was not a derivation.

Line 363, reference for using 6 K km-1 to indicate stable layers?

No reference is needed. This was the threshold we imposed purely for the purpose of highlighting the highest regions of stability within the profiles. It is not meant to be universal; it exceeds the largest theta gradient of the wet adiabat in this instance (See Fig 4c) but otherwise carries no special significance as a threshold.

Figure 6, wouldn't it be nicer to be able to see LCL (or MLH) directly from the graph? (also check panel (1) for misprinting).

Thank you for catching the misprint. This has been resolved.

Line 499-503, not sure if I follow this argument, how does spatial scale come in to play here, may need to clarify.

Thank you for raising this question. If the variability in question is smaller than the scale of the measurement (sampling duration/interval times flight speed) undilute "pockets" would not be identifiable within a larger bulk average.

A minor text revision has been added to clarify that 1 s corresponds to the sampling interval and <100 m corresponds to the equivalent spatial scale.

Discussion and Conclusions, you already touched on this by providing a bullet list of interests for future investigations, but I think it's worth emphasizing/mentioning the strengths of this dataset (i.e., rich in detail and high spatiotemporal resolution that will shift the focus away from statistical description of features in an ensemble sense) in the context of stimulating future topic-oriented studies.

Agreed we have added a further concluding remark to highlight the benefit of the combination of the unique ACTIVATE dataset with this targeted sampling strategy.

"The datasets collected during ACTIVATE provide a unique and comprehensive characterization of dynamic, thermodynamic, trace gas, aerosol and cloud properties at high temporal resolution, and the implementation of this flight strategy provides targeted observations of individual cloud clusters and their environments."

Reviewer 2:

The manuscript "*Measurement Report: Cloud and environmental properties associated with aggregated shallow marine cumulus and cumulus congestus*" by Crosbie et al provides an insight into the measurements of six case-studies during the ACTIVATE field campaign from 2020 to 2022, and thereby aims to characterize the cloudiness and its environment by means of their thermodynamic, kinematic and microphysical properties. The measurements (supplemented by some reanalyses) are from two aircraft – a King Air primarily taking on the remote-sensing responsibilities with dropsondes and a Falcon jet flying in and out of clouds to get in-situ measurements. The value of the coordinated strategy of these flights is reflected in the possible synergies of the measurements that are highlighted here in various aspects. The manuscript effectively describes the cloud and environmental quantities of the six case-studies and can be a great starting point for subsequent investigations that wish to go in to much greater depth in understanding processes that link marine clouds and their environment.

I found the manuscript well-written, aptly structured, and clear in its presentation almost everywhere except for the quality of the figures. Here, I list some general comments (mostly minor) that I believe can add value to the existing state of the manuscript and request clarifications on some aspects.

**General comments**

1. Motivation for the *observations*: The current introduction is comprehensive and presents the complexity in processes with respect to scales and quantities. However, it leaves me hanging without a more direct link (or more specific questions) from the processes to the observations being discussed here. Are there observational challenges that have restricted us from getting a better picture? The manuscript needs to motivate the reader as to which are the unobserved processes, quantities or hitherto missed synergies that are accomplished here which will help answer specific questions of mesoscale aggregation that the introduction section raises. These observational "gaps" in the context of the mentioned process-understanding problems will motivate why such a synergy of two aircraft spanning in-situ and remote-sensing measurements is valuable and which strategies and measurements will subsequently help in improving the current state of process understanding.

It is very challenging to construct a satisfactory response to this comment because it is very difficult to discuss observational challenges and missed synergies at this stage. The main intent of this paper was to document a series of measurements that are intended to promote detailed study of the processes and sub-processes outlined in the introduction. We felt it necessary to outline these detailed processes as motivation for our sampling method, but it is premature to make a concrete determination of the specific drivers from mesoscale aggregation in the context of observed/unobserved processes and quantities.

Hopefully after several of those intended follow-on studies are completed, we may be in a better place to comment on the specific value of the set (or subset) of measurements that provide the most impact. The advantage of a comprehensive observational dataset is that understudied processes can now be investigated. However, many of the quantities outlined here may have little controlling impact on mesoscale cloud aggregation but it does not diminish their value within a comprehensive dataset. Of equal value, we may discover that targeted flights around cloud clusters such as these offer useful natural laboratory conditions for studying cloud-environment interactions that are otherwise challenging to untangle from a more statistical sampling approach.

2. In Section-7, there is not much mention about how these quantities are associated with the cloud processes, e.g. their lifecycle stage and other properties such as ql, Nd, etc, except a brief mention in L712-713. I realise it is out of the scope of this paper to characterize the subprocesses of the cases in detail, but in its current state, the section seems isolated from the discussions in the previous sections. If the authors could outline (or hypothesize) how these quantities might be related physically (or which physical sub-processes might be dominant so as to show such measurements), that would be quite useful for any subsequent investigations that will use these data. I see these have been sparsely mentioned in the conclusion, but some details in Section-7 would be good.

Respectfully, we are comfortable with not making too many additional speculative connections between the environmental aerosol microphysics and the cloud processes in this report. The connections that are already made are the ones we wished to highlight, namely, (i) the closure of cloud base activation, (ii) the potential of entrained aerosol at higher levels as a source for secondary droplet activation, (iii) the connections between the aerosol data described in Section 7 and Section 6 (composition). We have made a minor addition to the text to highlight another link back to Section 6.

The aerosol data were presented in this way with future LES modelling in mind because size distributions and the vertical profile of number concentration sufficiently capture the aerosol environment for, say, an aerosol-cloud interaction study. We agree with the reviewer that it is mostly out of scope to dive into the sub-processes too much here. Like other sections, Section 7 documents the measurements with future sub-process investigations in mind. However, we did not find a compelling need to belabor the discussion here when these interactions may be of secondary or tertiary importance for cloud aggregation processes.

In this report there have been a few places where we have made hypothetical judgements on the measurements, but generally we have tried not to venture too far into making those types of assessments. It is our understanding that this is one of the factors differentiating an ACP measurement report from a research article. Some of these areas are currently being worked on as part of dedicated studies by others (e.g., there is a dedicated study about aerosol vertical profile behavior near Bermuda).

3. The study does not make any connections to similar measurements in existing literature or similar studies from other recent campaigns* in the region. It would be good to point out which aspects of shallow convection previously observed were confirmed by these case-studies, and/or what was new in these ACTIVATE cases. For example, the statements in L324 and L362 about moisture and stability is similar to observed case-studies from the NARVAL2 campaign. Connect these observations with similar strategies such as in the EUREC4A campaign, where the core measurement strategy also involved one aircraft (mainly remote sensing) above and another below (mainly in-situ).

The reviewer makes a valid point regarding the George paper in relation to the moisture and stability finding. We have added this reference to that section. We have been a bit cautious to try and make too

many presumptive links with conclusions drawn from the shallower more capped winter trade conditions, but this one is certainly useful.

Beyond that specific example, it is not clear why we are required to explain the connection with these other example strategies. Within these examples, there are clearly overlapping themes but dissecting the details are probably not within the scope of the measurement report.

4. Related to the point above, it feels like a missed opportunity with so many dropsonde launches that no estimation of the area-averaged measurements such as vertical velocity (W) first shown by Bony and Stevens (2019) and carried out extensively in campaigns such as EUREC4A and OTREC. This measurement provides a very important characterization of the environment. Although the spatial scale and time-advection would be different for these 6 case-studies, it provides more data points to compare with efforts of other cloud-organization studies that have shown W to be an important environmental factor (e.g. recently by Vogel et al, 2022)

First as a minor point of clarification, determining W is not simply an area-averaged quantity it is a gradient quantity that is then integrated vertically. While the method attempts to predict the area-average, the difference in magnitudes between averages and anomalies makes that fact an important distinction for sparsely sampled data like these collections of dropsondes.

The averaged quantities that we did report (temperature, water vapor, horizontal wind components and the vertical gradients of the horizontal winds i.e., shear) are likely representative of the area average because of the near uniformity of the dropsondes and the possibility of sample bias is low because the mean is large and the perturbations relatively small. The W method relies on a lack of bias in the sampling of the perturbations. These cases are too complex to make that a blanket assumption without more careful analysis. As noted, there is evolving deep convection within the relative vicinity as well as quiescent regions and other nearby clusters of similar magnitude. Equally important is the consideration of the advection and evolution of the mesoscale W relative to the spatial and temporal scale of the dropsonde array. This may be small enough to ignore elsewhere but not here.

With no doubt, we are in full agreement of the immense value provided by a characterization of the mesoscale circulation associated with these cases; however, it is not a missed opportunity. That analysis is deserving of its own study, particularly since these derived W values are not readily verifiable by other means.

\* Examples of process case-studies in other campaigns:

- **OTREC** (Raymond, D. J., & Fuchs-Stone, Ž. (2021). Emergent properties of convection in OTREC and PREDICT. Journal of Geophysical Research: Atmospheres, 126, e2020JD033585. https://doi.org/10.1029/2020JD033585)

- **NARVAL2** (George, G., B. Stevens, S. Bony, M. Klingebiel, and R. Vogel, 2021: Observed Impact of Mesoscale Vertical Motion on Cloudiness. J. Atmos. Sci., 78, 2413–2427, https://doi.org/10.1175/JAS-D-20-0335.1.)

- **EUREC4A** (Touzé-Peiffer, L., R. Vogel, and N. Rochetin, 2022: Cold Pools Observed during EUREC4A: Detection and Characterization from Atmospheric Soundings. J. Appl. Meteor. Climatol., 61, 593–610, https://doi.org/10.1175/JAMC-D-21-0048.1.)

**Specific comments / clarifications**

- All figures: Please vectorize for better quality and accessibility while zooming in.

The figure quality has been improved. The issue is believed to be a pdf conversion issue.

- Make Section-2 title more suited to the study, e.g. "Measurement strategies and Data"

Thanks. This was already changed in accordance with the other reviewer's suggestion.

- In Section-3, mention the differences in the diel placement of the sampling between the 6 cases. From Table-1, it seems like Cases 1 & 6 (~15-16 UTC) are at a similar timeframe while the rest are all at another but not too varying timeframe (1830 - 2000 UTC). It's good to mention the sampling time difference among the cases, and if any quantities showed expected/unexpected deviations from the diel cycle of convection (e.g. Vial et al, 2019).

Thank you - this is a useful addition. However, we felt that the more appropriate place to list the times of the flights was at the end of Section 2.1 rather than Section 3, which currently only summarizes the synoptic meteorology not the specifics of the flight operation. Since the relationship with solar noon is probably the more influential aspect, we have included that information too. We share the reviewer's interpretation that the timeframes are not overly varying so did not want to wrongly suggest the value of this dataset for evaluating diel effects.

"Cases 1 and 6 occurred earlier in the day with module midpoints approximately an hour prior to local solar noon (15-16 UTC) while Cases 2-5 were later occurring approximately 2-4 hours after local solar noon (18-20 UTC) as a result of being the second flight of a two-flight day."

- L285-286: It is not clear how the cloud features were selected and how far in advance was this selection made. This would also clarify what "by design" means.

The design of the module is schematically illustrated in Figure 1 and was used as a template for the folks involved in executing the flight plan. By design, the sampling area has clouds in the center of it because that was the scenario that was sought.

Our including "by design" was merely to acknowledge the obviousness of the satellite matching the picture we visually sought from the aircraft but since it caused confusion, we have now removed it.

Other than to state that Figure 1 was the template for picking cloud features, we can only add to that by providing a narrative for how this was done in practice. However, we elected not to dive into a description of the mechanics of this process in the paper and we stand by that decision. The question of how far in advance is multifaceted. There was a component involving assessing broad regions of interest (and feasibility) which was usually 2-4 hours prior, a component involving real time tracking with geostationary satellite usually once the two aircraft were airborne and heading toward the region of interest usually involving a shortlist of 2-3 coordinates sent to the Falcon. Then the scientist on the Falcon made a final determination and the center point coordinate was communicated to the King Air. Each case was different and involved considerable real time decision making. Some other candidate cases were attempted on other flights but were abandoned because they did not materialize (not reported here).

- L302: why not Case-3 (6.7 h)?

Thank you for catching that. The intent of the sentence was to highlight the fact that the true lifetime of Case 1 and 2 was hard to judge because of the other complicating factors. A minor adjustment has been made to remove the apparent contradiction.

- L350: You could point to existing literature, where Mieslinger et al 2022 show the sensitivity of lidars can capture optically thin clouds that are often missed by coarse-resolution satellites.

Excellent, thank you.

- L351: Were there any large variabilities among the dropsonde profiles of a single case? e.g. due to cold-pools such as what Touzé-Peiffer et al, 2022 show? Also in L366: Are there variabilities among dropsondes in stability for launches within cloudy and clear areas?

Yes, there was variability amongst the dropsonde ensemble but the moisture and temperature relationship to cloudy/clear zones was not as intuitive as might be anticipated. After much consideration we decided not to include details of the intra-case sonde variability here because along similar lines to the response to General Comment 4, these topics would require a large volume of explanation and detailed analysis to address them with the requisite attention they deserve.

- L400-403: This could have been interesting to see from the perspective of area-averaged W (mentioned in general comment-4)

We would urge caution in conflating these two aspects. The description of convective velocities and the growth and decay of thermals is happening at quite a rapid turnover rate. Even though the in situ characterization of the clouds was taking place near the geometric center of a selected cluster, within that region there could be several active cells at different lifecycle stages and within the area spanned by the dropsondes there were often additional clusters. Therefore, it would be difficult to attribute any linkage between the periodic/pulsating growth/decay of thermals comprising the main cluster with a change in the larger circulation assessed through the dropsonde W.

- L448: Doesn't this counter the statement made in L442 that mean Nd does not change over the lifecycle?

Thank you for catching that. It was not our intention to imply a generalized statement regarding the lifecycle. Now that it has been pointed out, it is clear that, as written, an unintended connection may be implied. We have made alterations to remove the ambiguity. The underlying causes behind the decaying Case 5A (a very small system lacking forcing/support and succumbing to entrainment desiccation) versus the mature congestus of Case 4A are undoubtedly different so it should not be expected that the factors influencing the microphysical changes in one be consistent with the other.

A statement to this effect was added:

"The contrasting $N_d$ behavior seen in Case 5 compared to the consistent $N_d$ characteristics of Case 4 highlight the different role of aerosol interactions across lifecycle for different cloud system types."

- L470: It isn't clear why data should coalesce around (1,1) especially if most points are away from the adiabat.

It is now clear that perhaps this is misleading. Most of the data points are at much lower adiabatic ratios (as stated) but the point here was to show that it tends towards (1,1). Setting the adiabatic $q_L$ has uncertainty based on the height of the LCL and the $N_{d0}$ estimate was only as good as our ability to find points that we could classify as being close to adiabatic. To clarify we have changed "coalesce" to "terminate".

- L670-673: How does this reconcile with the increased presence of the ammonium ion in AMS mass concentration? Did they not react with sea-salt much in Case-1? Or did they come in from upper layers seeing as it seems fairly top-heavy in Case-1 compared to the well-distributed Cases-2 and 3?

The reviewer's question is somewhat unclear. We acknowledge they are noting the difference in the submicron aerosol relative mass fraction profile of $NH_4^+$ in Case 1 compared to 2 and 3, which is more enhanced at higher levels during Case 1. However, the comment appears to reference text discussing the cloud water composition.

In all cases, fine mode aerosol nitrate is small. The discussion in the text is about the enhancement of cloud water nitrate in Cases 2 and 3 over Case 1. Nitrate in cloud water may arise from gas phase nitric acid, fine mode nitrate (not really in this instance) or coarse mode usually attached to sea salt or dust. The discussion was mainly highlighting the possible connection between the sea salt in Case 1 and clean marine HSRL typing and sea salt in Cases 2 and 3 with more polluted marine typing. The composition measurements are not size resolved, nor are there gas phase measurements (relevant to this) so it is difficult to definitively apportion the sources of cloud water species.

- L688: Please specify what ranges the Aitken and accumulation modes usually occupy and their relative importance in cloud processes.

No. Labelling these modes as Aitken and accumulation is only to follow convention. The important values of these modes are the values provided in the table. The Hoppel reference is a suitable explanation for their role in cloud processes.

- L784: How does the cloud situation compare with other studies that have looked at cases with Saharan dust in the environments (e.g. Gutleben et al, 2019)?

These are concluding/summarizing remarks highlighting the similarities and differences amongst the cases, so we do not wish to discuss this connection here. Instead, we have added the following to Section 6.2:

"Long-range transport of Saharan dust has been shown to inhibit cloud growth in this region (Gutleben et al., 2019) and may partially explain some of the characteristics here (e.g., the differences between Case 5A and Case 5B)."

- Figure-2: Difficult to distinguish the different contour lines. Could the plot be made clearer somehow? Or the caption more detailed?

We are attempting to condense a lot of meteorological data onto single plots because overlaying the data provides the best way to interpret the patterns. On 2d, for example, we employed the standard way of presenting an upper air chart 500 mb geopotential lines overlaid with thickness. The only exception in the thickness presentation was that we changed the threshold for transition to 564 dm rather than the standard 540 dm because it better matched the subtropical temperatures.

Yes, we appreciate that these plots are complex, but they intentionally contain a lot of information.

- Figure 3d: Please consider changing the color. It is tough to see between the clouds and pink line that there is an "x" there.

The X marking the location of Case 4B at the time of the image has been replaced with a square.

- Figure-5: How is the principal axis determined? Is it an approximation of the cloud cluster object to an ellipse? If so, how is the object defined?

Nothing sophisticated, just an estimate of the main orientation of cloud organization including nearby features. We have changed "principal" to "primary" and "assessed" to "estimated" to avoid any anticipation that this was more sophisticated than it really was.